# Autocorrelation Matters: Understanding the Role of Initialization Schemes for State Space Models

**Fusheng Liu, Qianxiao Li**
National University of Singapore, Singapore
`fusheng@u.nus.edu, qianxiao@nus.edu.sg`

## Abstract

Current methods for initializing state space model (SSM) parameters mainly rely on the HiPPO framework (Gu et al., 2023), which is based on online function approximation with the SSM kernel basis. However, the HiPPO framework does not explicitly account for the effects of temporal structures of input sequences on the optimization of SSMs. In this paper, we take a further step to investigate the roles of SSM initialization schemes by considering the autocorrelation of input sequences. We: (1) rigorously characterize the dependency of the SSM timescale on sequence length based on sequence autocorrelation; (2) find that with a proper timescale, allowing a zero real part for the eigenvalues of the SSM state matrix mitigates the curse of memory while still maintaining stability at initialization; (3) show that the imaginary part of the eigenvalues of the SSM state matrix determines the conditioning of SSM optimization problems, and uncover an approximation-estimation tradeoff when training SSMs with a specific class of target functions.

## 1 Introduction

The state space model (SSM) is a sequence model that has recently shown great potential in long sequence modeling across various applications, including computer vision (Zhu et al., 2024; Liu et al., 2024), time series forecasting (Rangapuram et al., 2018; Zhang et al., 2023) and natural language processing (Gu & Dao, 2023; Dao & Gu, 2024). In mathematics, a SSM layer is defined by a continuous-time ordinary differential equation $h'(t) = Wh(t) + Bx(t)$, $y(t) = Ch(t) + Dx(t)$, where $W, B, C, D$ are trainable parameters, $x(t)$ is the input sequence, and $y(t)$ is the output sequence. For discrete input sequences, a timescale $\Delta > 0$ will be introduced as a hyperparameter to discretize the model. Different from the attention mechanism (Vaswani et al., 2017), SSMs are recurrent-based architectures that treat the input sequence token by token, yet can achieve first-order time complexity on the sequence length through parallelization (Gu et al., 2022b). There are two well known issues for training recurrent-based architectures, the vanishing and the exploding gradient problems (Pascanu et al., 2013). By introducing complex-valued initialization schemes, proper parameterization methods and regularization techniques, recent works demonstrate that SSMs can achieve performance comparable to attention-based architectures in terms of both computational cost and sample efficiency (Gu & Dao, 2023; Dao & Gu, 2024; Zhu et al., 2024; Yu et al., 2024; Wang & Li, 2024; Liu & Li, 2024; Yu et al., 2024; Bick et al., 2024; Hwang et al., 2024; Wang et al., 2024; Waleffe et al., 2024). However, the theoretical understanding on the roles of the initialization schemes is still lacking and needs to further explored. In this paper, we particularly look into the timescale $\Delta$ and the state matrix $W$, and we aim to study the following fundamental question

*Given a sequential dataset with length $L$, how should the timescale $\Delta$ depend on $L$ and what is the role of $W$ on training SSMs?*

Based on the analysis of continuous-time SSMs, previous works (Gu et al., 2022b;c; 2023) propose the HiPPO framework where $W, B$ are initialized such that the SSM basis kernels $\{e_n^\top e^{Wt}B\}_{n=1}^\infty$ are orthogonal in $L^2[0,\infty)$ with some measure $\omega(t)$, and the timescale $\Delta$ scales as $1/L$ to capture long range dependencies of sequences with length $L$. Common HiPPO-based initialization methods such as S4D-Legs and S4D-Lin typically presume that the measure $\omega(t)$ is exponential decay and

the discrete input sequences $x$ have a inherent timescale $\Delta$ that is shared with the model. However, these assumptions are restrictive because exponential decay measures weaken the effects of temporal dependencies in input sequences, and in practice, we usually lack prior information about the data's timescale. To address this concern, we take an initial step towards understanding the relationship between the autocorrelation of input sequences and the SSM initialization schemes. Specifically, we focus on the diagonal SSM[1] (Gu et al., 2022c) where the state matrix $W$ is a complex-valued diagonal matrix. By studying the stability condition for given input sequences $x \in \mathbb{R}^L$, we find that the connection of the timescale $\Delta$ and the sequence length $L$ is highly related with the spectrum of the data autocorrelation matrix $\mathbb{E}[xx^\top]$. Different temporal dependencies in the input sequences can cause significant variations in the spectrum of the autocorrelation matrix. For example, when $x$ is sampled from a standard normal distribution, $x$ has zero temporal dependencies, and the autocorrelation matrix becomes an identity matrix. On the other hand, if $x$ consists of constant values, the input sequence exhibits full temporal dependencies, and the autocorrelation matrix is low rank. For the state matrix $W$, our stability analysis shows that even with a zero real part, i.e $\Re(W) = 0$, the diagonal SSM can still be stable at initialization if $\Delta$ is properly set. One benefit for setting the real part to zero is that the learned SSM kernel functions at initialization do not exponentially decay, which helps to mitigate the curse of memory (Li et al., 2022). Our convergence analysis indicates that the imaginary part $\Im(W)$ plays a crucial role in the convergence rate and explains the benefits for complex-valued SSMs compared to real-valued SSMs in terms of the optimization. In particular, the more separated the imaginary parts $\Im(w)$ are, the faster the convergence. When considering both approximation and optimization, we characterize an approximation-estimation tradeoff when the target function has closely spaced dominant frequencies. Then well separated $\Im(w)$ values lead to fast convergence, while achieving a good approximation requires close imaginary parts.

To summarize, the main goal of this paper is to provide a theoretical understanding on the effects of three specific hyperparameters: the model timescale $\Delta$, the real part of $\Re(W)$, and the imaginary part of the state vector $\Im(W)$. These components are connected as a *data-dependent* initialization scheme for SSMs. First, for any given sequential dataset, we can estimate its autocorrelation. Using this information, we can apply Theorem 3.1 to initialize $\Delta$, taking into account both data autocorrelation and sequence length. Second, if the true input-output mapping is represented by an underlying linear functional, often referred to as a memory function, that exhibits a long memory pattern, our second theory, detailed in Section 3.2, suggests that initializing with a zero real part can help mitigate the challenges posed by long sequences. Finally, the third theory introduced in Section 3.3 discusses an approximation-estimation tradeoff that arises when the true memory function $\rho^*$ features closely spaced frequencies. If we can accurately recover $\rho^*$ from the sequential data, we can then initialize the imaginary part based on the dominant frequencies of $\rho^*$, thereby finding an optimal balance informed by theoretical insights. Accordingly, our contributions are as follows:

- In section 3.1, we characterize the dependency between the timescale $\Delta$ and the sequence length $L$ by taking into account the autocorrelation of the input sequences. Even if the eigenvalues of the state matrix $W$ have zero real part, the stability condition on the magnitude of the output value at initialization can still hold with an appropriate setting of $\Delta$.

- In section 3.2, we show that the real part of the eigenvalues of the state matrix $W$ determines the decay rate of the SSM kernel functions. Allowing the eigenvalues of $W$ to have zero real part at initialization can significantly increase the model's effective memory and help mitigate the curse of memory for fixed-length tasks that require long-term memory.

- In section 3.3, we prove that the conditioning of SSM optimization problems is determined by the separation distance of the imaginary parts of the eigenvalues of the state matrix. Well-separated imaginary parts induce faster convergence, whereas closely spaced ones lead to slower convergence. This explains the benefits of complex-valued SSMs over real-valued SSMs. Furthermore, it uncovers an approximation-estimation tradeoff when the target function has close dominant frequencies in the frequency domain.

## 2 PRELIMINARIES

In this section, we briefly introduce the diagonal SSM and the problem setting we consider throughout this paper. Specifically, we consider the following single-input single-output (SISO) diagonal-

---

[1]To simplify the analysis, we omit the skip connection by letting $D = 0$.

SSM built in the complex number field $\mathbb{C}$ and then cast into the real number field $\mathbb{R}$,

$$\frac{d}{dt}h(t) = Wh(t) + bx(t), \quad y(t) = \Re(c^\top h(t)), \quad t \geq 0, \tag{1}$$

where $\Re(\cdot)$ represents the real part; $x(t)$ is input sequence from an input space[2] $\mathcal{X} := C_0(\mathbb{R}_{\geq 0}, \mathbb{R})$; $y(t) \in \mathbb{R}$ is the output sequence at time $t$; $h(t) \in \mathbb{C}^m$ is the hidden state with $h(0) = 0$; $W \in \mathbb{C}^{m \times m}, b, c \in \mathbb{C}^m$ are trainable parameters. In particular, the state matrix $W = \text{diag}(w_1, \ldots, w_m)$ is a diagonal matrix. To simplify the analysis, we omit the skip connection matrix $D$. Following the training setup in Gu et al. (2022c), the read-out vector $c$ follows standard normal distribution and the read-in vector $b$ in (1) is fixed as an all-one vector at initialization without training. Under these settings, the input-output relation in (1) is explicitly given by the integral

$$y(t) = \int_0^t \Re(c^\top e^{ws}) x(t-s) ds, \tag{2}$$

where $w \in \mathbb{C}^m$ is the state vector that contains all the diagonal entries of the state matrix $W$, and the function $\Re(c^\top e^{ws})$ is called the memory function or the kernel function.

**Discretization.** To handle discrete sequences, we follow (Gu et al., 2022c) to use the zero-order (ZOH) hold method for discretization. Then given a timescale $\Delta > 0$ and any discrete sequence $(x_0, \ldots, x_{L-1}) \subset \mathbb{R}$ with length $L$, for $\ell = 1, 2, \ldots, L$, the ZOH method induces a model output

$$y_\ell = \Re\left(\sum_{j=1}^m \frac{e^{\Delta w_j} - 1}{w_j} c_j e^{\Delta w_j(\ell-1)}\right) x_0 + \cdots + \Re\left(\sum_{j=1}^m \frac{e^{\Delta w_j} - 1}{w_j} c_j e^{\Delta w_j 0}\right) x_{\ell-1}. \tag{3}$$

*Remark* 2.1. Here we focus on the SISO model with ZOH discretization. It is also possible to extend to the multiple-input multiple-output (MIMO) case, by noticing that the MIMO output is essentially a linear combination of several single-input single-output (SISO) models. As a result, we can extend our results to the MIMO scenario by examining each SISO model individually along with its respective input-output mapping. However, our theory are not directly applicable to other discretization methods (e.g. bilinear method), which involve different matrix forms for the model's input-output mapping (see Appendix D), and require different techniques to yield theoretical insights.

In the following section, we tackle the problems related to the initialization schemes of SSMs that were introduced in the Introduction. Specifically, we will explore the following questions:

1. **Timescale Initialization**: How should we initialize the model timescale $\Delta$ for fixed-length tasks to enhance the training of SSMs? Is the previously used scaling $\Delta = 1/L$ a universal approach?

2. **Real Part of the State Vector**: What role does $\Re(w)$ play? Can we initialize $\Re(w)$ to be zero, and what benefits might arise from a zero real part?

3. **Imaginary Part of the State Vector**: What role does $\Im(w)$ play? What advantages do complex-valued SSMs offer compared to real-valued SSMs?

## 3 MAIN RESULTS

In this section, we present our main results by focusing on three initialization parameters $\Delta, \Re(w)$ and $\Im(w)$ respectively. Specifically, in section 3.1, we rigorously characterize the relationship between the timescale $\Delta$ and the sequence length $L$ in terms of training stability at initialization by taking into account data autocorrelation. In section 3.2, we demonstrate that allowing the state vector's real part to be zero can prevent exponential decay in the SSM kernel function, thereby mitigating the curse of memory in certain scenarios. In section 3.3, we explore the relationship between the convergence rate and the separation distance of the state vector's imaginary part. In particular, we uncover an approximation-estimation tradeoff for a class of target functions.

### 3.1 RELATIONSHIP BETWEEN $\Delta$ AND $L$

In this subsection, we derive a stability condition for the ZOH-discretized diagonal SSM (3) when the state vector's real part $\Re(w)$ is non-positive. From both theoretical and numerical perspectives,

---

[2] A linear space of continuous functions from $\mathbb{R}_{\geq 0}$ to $\mathbb{R}$ that vanishes at infinity.

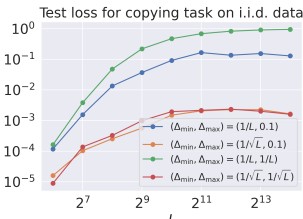 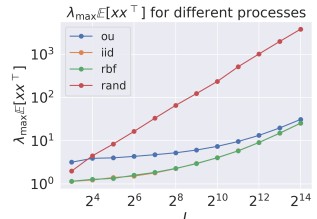 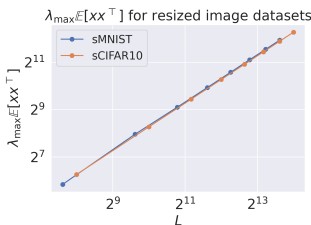

Figure 1: (Left) Training a diagonal SSM (3) on a copying task using i.i.d. data with a dimension of 128. We vary the minimal timescale $\Delta_{\min} = 1/L, 1/\sqrt{L}$ and the maximal timescale $\Delta_{\max} = 1/L, 1/\sqrt{L}, 0.1$ w.r.t. sequence length $L$. (Middle) The maximal eigenvalue of the autocorrelation matrix $\mathbb{E}[xx^\top]$ on different random processes of $x$. (Right) The maximal eigenvalue of $\mathbb{E}[xx^\top]$ on sequential image datasets sMNIST and sCIFAR10 with different resize rates varied from 0.5 to 4.

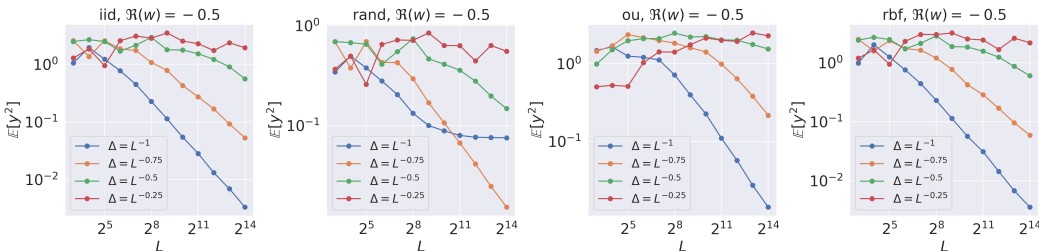

Figure 2: The expected magnitude of the SSM output value on synthetic sequences with different autocorrelation. The real part $\Re(w) = -0.5$ follows the common practice and we consider four dependencies between the timescale $\Delta$ and the sequence length $L$.

we demonstrate that the dependency of the model timescale $\Delta$ on the sequence length $L$ is strongly influenced by the data autocorrelation. To start with, we prove the following theorem that provides an upper bound on the magnitude of the model output value.

**Theorem 3.1.** *Consider a ZOH discretized SSM (3) with timescale $\Delta > 0$ and $\Re(w_j) \leq 0$ for $j = 1, \ldots, m$. Suppose that the input sequence $(x_0, \ldots, x_{L-1})$ is sampled from a unknown distribution in $\mathbb{R}^L$, and the read-out vector $c$ is from i.i.d. standard normal distribution. Then we have*

$$\mathbb{E}_{c,x}[y_L^2] \leq \Delta^2 m^2 L \cdot \lambda_{\max}(\mathbb{E}[xx^\top]),$$

*where $\lambda_{\max}(\cdot)$ represents the maximal eigenvalue.*

The proof is provided in Section D. In practice, the hidden state size $m$ is often much smaller than the sequence length $L$ (Gu et al., 2023). Given this, we focus on fixing the hidden size $m$ and investigating the relationship between the model timescale $\Delta$ and the sequence length $L$. We see that Theorem 3.1 connects the model timescale $\Delta$ with the sequence length $L$ in terms of the data autocorrelation matrix $\mathbb{E}[xx^\top]$. If we have normalized the sequences such that $\mathbb{E}[\|x\|^2] = 1$, then a simple observation is that $1 \leq \lambda_{\max}(\mathbb{E}[xx^\top]) \leq L$ because $\text{Tr}(\mathbb{E}[xx^\top]) = L$. This indicates that the maximal eigenvalue of the autocorrelation matrix can have different dependencies on $L$ based on the temporal dependencies. For example, when the elements in the sequence are uncorrelated with each other, $x$ exhibits *zero* temporal dependencies, and the autocorrelation matrix is an identity matrix with $\lambda_{\max}(\mathbb{E}[xx^\top]) = 1$. In this case, $\Delta$ should scale as $1/\sqrt{L}$ to ensure training stability. On the other hand, when $x$ is a constant sequence $(1, 1, \ldots, 1)$, then $x$ exhibits *full* temporal dependencies. The autocorrelation matrix then becomes a rank-1 matrix with $\lambda_{\max}(\mathbb{E}[xx^\top]) = L$, implying that $\Delta$ should scale as $1/L$. Additionally, this upper bound is applicable for all cases where $\Re(w) \leq 0$. As the real part $\Re(w)$ approaches zero, the exponential decay rate of the SSM kernel slows, resulting in a tighter bound. Specifically, when the real part is zero and the input data lacks temporal dependency, the bound becomes tight up to a constant factor. This occurs because the data autocorrelation matrix $\mathbb{E}[xx^\top]$ simplifies to a diagonal matrix under these conditions. Given a specific task with an input sequential dataset, we can then initialize the model timescale $\Delta$ as $\mathcal{O}(1/\sqrt{L\lambda_{\max}(\mathbb{E}[xx^\top])})$.

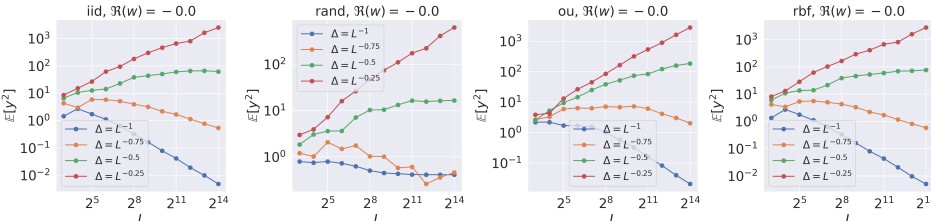

Figure 3: The expected magnitude of the SSM output value on synthetic sequences with different autocorrelation and different dependencies between $\Delta$ and $L$. The real part $\Re(w)$ is set to be zero.

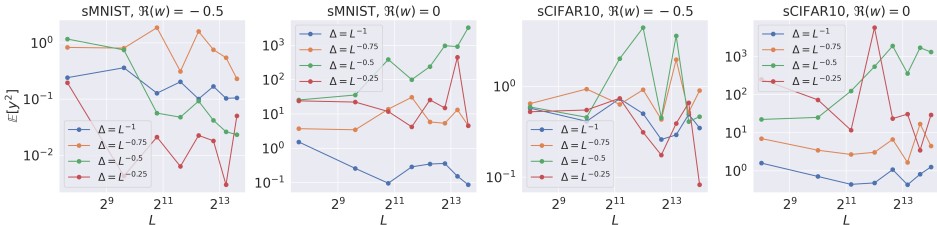

Figure 4: The expected magnitude of the SSM output value on sequential image datasets with different resize rates (ranging from $0.5$ to $4$) and different dependencies between $\Delta$ and $L$.

*Remark* 3.2. Theorem 3.1 applies for the final output mode, while in practice, there are also some other output modes and the analysis on the stability condition is case by case. For example, for the pooling mode $y = \frac{1}{L} \sum_{\ell=1}^{L} y_L^2$, we can use Theorem 3.1 to get a same upper bound for $\mathbb{E}_{c,x}[y^2]$. Also, in this paper we consider fixed-length tasks, i.e., all the sequences are with the same length. For the varied-length case, we may first cluster the sequences into several groups, and then increase the model feature dimension to manage varied sequence lengths separately. For example, if the sequence length alternates between $L_1$ and $L_2$, then we can double the feature dimension and initialize the model separately for the first and second halves, allowing the model to accommodate two fixed-length datasets simultaneously.

**Numerical experiments on $\lambda_{\max}(\mathbb{E}[xx^\top])$ and $\mathbb{E}[y_L^2]$.** To validate our theory, we conduct experiments on the exact values of the magnitude of the model output and $\mathbb{E}[xx^\top]$. Specifically, we consider both synthetic and real sequential datasets in both negative and zero real part cases. For synthetic datasets, we consider Gaussian process with mean $0$ and autocovariance function $\mathbb{E}[x_i x_j] = K(i,j)$. By restricting $K(i,i) = 1$ then the autocovariance matrix is exactly the same as the autocorrelation matrix. In this paper, we choose $4$ Gaussian processes with different autocovariance functions and plot their maximal eigenvalues. The autocovariance functions for "ou, iid, rbf" are $K(i,j) = \exp(-|i-j|/\ell), \delta_{i-j}, \exp(-|i-j|^2/\ell)$ respectively. The autocovariance matrix for "rand" is given by $\Sigma \Sigma^\top$ where $\Sigma$ is a random matrix with i.i.d. uniform distributed entries in $[0,1]$. As Figure 1 (Middle) shows, different processes have varying dependencies of $\lambda_{\max}(\mathbb{E}[xx^\top])$ on $L$ ranging from $\mathcal{O}(1)$ to $\mathcal{O}(L)$. For the i.i.d. case, $\lambda_{\max}(\mathbb{E}[xx^\top])$ is not always $1$ in Figure 1 (Middle), which is because we use the sample autocorrelation matrix to replace the expected autocorrelation matrix. For real sequential datasets, we choose to resize the MNIST dataset (LeCun et al., 2010) and the gray CIFAR10 dataset (Tay et al., 2021) with resize rates $[0.5, 1, 1.5, 2, 2.5, 3, 3.5, 4]$ and the flatten the images to sequences. More experiment details are provided in Appendix B. We record $\lambda_{\max}(\mathbb{E}[xx^\top]$ based on the entire training dataset. As shown in Figure 1 (Left), the maximal eigenvalue scales (almost) linearly with sequence length across the resize rate for both sequential MNIST (sMNIST) and sequential CIFAR10 (sCIFAR10) datsets. Additionally, we plot the relationship between the magnitude of the model output value and sequence length by varying the timescale $\Delta = [L^{-1}, L^{-0.75}, L^{-0.5}, L^{-0.25}]$. In Figures 2 and 4, when $\Re(w) = -0.5$ (following the setup in Gu et al. (2022c; 2023)), the magnitude $\mathbb{E}[y_L^2]$ remains stable for both synthetic and resized image datasets for all decay rates of $\Delta$. When $\Re(w) = 0$, Figures 3 and 4 demonstrate that for the 'rand' process, $\Delta = L^{-1}$ is stable. For the 'iid,' 'ou,' and 'rbf' processes, $\Delta = L^{-0.75}$ is stable. This indicates that our bound in Theorem 3.1 effectively characterizes the relation between $\Delta$ and $L$ for

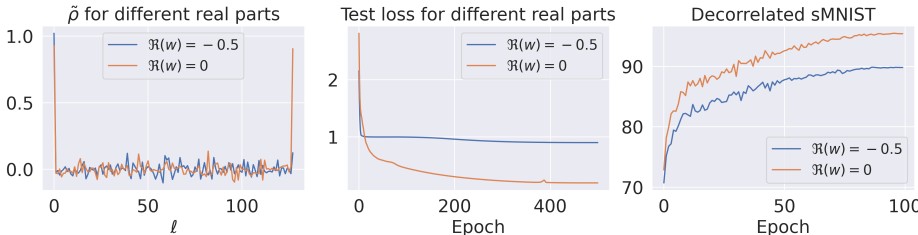

Figure 5: (Left) Training a diagonal SSM (3) on a task that requires long-term memory. The learned memory function $\tilde{\rho}$ effectively captures the spike in long-range dependencies. However, it struggles to do so when the real part is negative. (Middle) Test loss on the long-term memory task when initializing $\Re(w) = 0$ and $\Re(w) = -0.5$. (Right) Test accuracy for training a diagonal SSM on decorrelated sequential MNIST dataset with different real parts at initialization.

$\Re(w) = 0$. Moreover, as shown in Figure 4, for the sequential image datasets, $\Delta$ should scale as $1/L$ to ensure stability when $\Re(w) = 0$; otherwise, the magnitude increases with sequence length. This finding aligns with the empirical results in Gu et al. (2022c) that $\Delta$ should scale as $1/L$ to effectively capture the range of dependencies for length $L$. But their theoretical reasons are based on Fourier analysis of continuous-time SSMs and do not explicitly account for the data autocorrelation.

**Experiments on copying task with different timescales.** We tested the performance of the diagonal SSM (3) on a copying task with various dependencies of $\Delta$ on $L$. It is worth noting that, as discussed in Jelassi et al. (2024), SSMs struggle with the copying task because the model's state dimension needs to scale linearly with the sequence length to memorize all the input tokens. However, the limitation highlighted in Jelassi et al. (2024) pertains to the length generalization task—i.e., training an SSM with short sequences and then testing it on longer sequences will fail if the hidden size $m$ does not grow linearly with $L$. Here, we focus on a fixed-length task, where both training and test sequences have the same length. We find that, with an appropriately initialized timescale, SSMs can effectively handle the copying task even with a small state size. In this paper, we use a diagonal SSM with a fixed state size $m = 32$ to learn a copying task on i.i.d. data with a dimension of 128, and the timescale $\Delta \in \mathbb{R}^{128}$. We vary the minimal and maximal timescales $(\Delta_{\min}, \Delta_{\max})$ with different dependencies on $L$. From Figure 1 (Left), we see that the combination $(\Delta_{\min}, \Delta_{\max}) = (1/L, 0.1)$, which is commonly used in practice (Gu et al., 2022c; 2023) to train real datasets, consistently performs worse than setting $\Delta_{\min} = 1/\sqrt{L}$. This stable scaling is in line with our theoretical suggestions for i.i.d. data. Therefore, the data autocorrelation is very crucial for us to get a good initialization scale on the timescale. More experiment details are provided in Appendix B.

## 3.2 BENEFITS OF ZERO REAL PART

In this subsection, we investigate the benefits of initializing $\Re(w) = 0$ for tasks that require long-term memory. In previous works (Li et al., 2021; 2022), it is shown that recurrent-based models suffer from the curse of memory in both approximation and optimization when there is long-term memory in the target. For example, we consider using a diagonal SSM (3) to learn a input-out relationship given by a real-valued target function $\rho^*$ such that

$$y_\ell^* = \rho_{\ell-1}^* x_0 + \cdots + \rho_0^* x_{\ell-1}, \quad \ell = 1, 2, \ldots, L.$$

The objective function is given by the squared difference between the model output $y_L$ and the corresponding label $y_L^*$. Then in a special case when the input sequences have zero temporal dependencies with $\mathbb{E}[xx^\top] = \mathbb{I}_L$, the expected mean squared error is given by

$$\mathbb{E}[|y_L - y_L^*|^2] = \|\tilde{\rho} - \rho^*\|^2,$$

where $\tilde{\rho}$ is a vector $\left(\Re\left(\sum_{j=1}^m \frac{e^{\Delta w_j} - 1}{w_j} c_j e^{\Delta w_j 0}\right), \ldots, \Re\left(\sum_{j=1}^m \frac{e^{\Delta w_j} - 1}{w_j} c_j e^{\Delta w_j (L-1)}\right)\right)$ that represents the model's memory, and $\rho^* = (\rho_0^*, \ldots, \rho_{L-1}^*)$. Therefore, a well-trained SSM means that the model memory function matches with the target function, i.e.,

$$\Re\left(\sum_{j=1}^m \frac{e^{\Delta w_j} - 1}{w_j} c_j e^{\Delta w_j \ell}\right) = \rho_\ell^*, \quad \ell = 0, \ldots, L-1.$$

Figure 6: Recovering the memory function $\rho$ on the decorrelated sequential MNIST dataset by solving a linear equation $X * \rho = Y$, where $X \in \mathbb{R}^{N \times L}$ is the collected sequence matrix, $Y \in \mathbb{R}^{N \times 10}$ is the one-hot label matrix, and $*$ is the convolution operator. Then $\rho \in \mathbb{R}^{L \times 10}$ has 10 channels and we plot the scaled function $\sqrt{L}\rho$ each channel to show the underlying memory patterns.

Then we can see that the curse of memory happens when the target function $\rho^*$ has a sudden spike in a very long distance. For instance, consider a shifting task that requires mapping an input sequence $(x_0, \ldots, x_{L-1})$ to a shifted sequence $(0, \ldots, 0, x_0)$. In this task, the target $\rho^*$ is $(0, \ldots, 0, 1)$, which is challenging for an exponentially decaying SSM kernel $\tilde{\rho}$ to capture long-term memory when $\Re(w) < 0$. However, if we allow the real part to be zero at initialization, then $\tilde{\rho}$ does not undergo exponential decay. As a result, we can potentially avoid the curse of memory, even for long sequences, in this scenario. It is worth noting that in this paper, we do not consider a stable parameterization to ensure $\Re(w) \leq 0$ strictly during training. This approach means $\Re(w)$ is likely to become positive during training. Our goal is to allow the model to learn directly from the data without introducing new variables, such as reparameterization methods, which could complicate the analysis. Otherwise, it would be unclear whether the improvements are due to the zero real part or the introduced reparameterization method. Our experiments demonstrate that initializing with a zero real part still helps enhance training, even without a stable parameterization. This suggests that, despite the potential optimization stability challenges during training, a zero real part can be beneficial for training on certain tasks.

**Experiments on the benefits of zero real part.** To validate the effectiveness of having a zero real part, we conduct experiments on both synthetic and real datasets that require long-term memory. For the synthetic task, we use i.i.d. sequential data to easily visualize the expected error via the memory function. The goal is to learn an input-output mapping from $(x_0, \ldots, x_{L-1})$ to $x_0 + x_{L-1}$, which requires the model to memorize both the first and last token. The target memory function $\rho^*$ is $(1, 0, \ldots, 0, 1)$. In our setting, the sequence length $L$ is 128, and the hidden state size $m$ is 32. As shown in Figure 5 (Left) and (Middle), the SSM with a zero real part outperforms the case with a negative real part. It is evident that by initializing $\Re(w) = 0$, the learned memory function is able to capture long range dependencies. For the real-world task, we utilize the sequential MNIST (sMNIST) dataset. Before training, we preprocess the entire dataset with a linear transformation to decorrelate the training sequences, resulting in an autocorrelation matrix that is an identity matrix. We recover the underlying target memory function by solving a least square problem $\min_\rho \|X * \rho - Y\|_F^2$ where $X \in \mathbb{R}^{50000 \times 784}$ is the collected sequence matrix, $Y \in \mathbb{R}^{50000 \times 10}$ is the one-hot label matrix, and $*$ denotes the convolution operator. The recovered target memory function $\rho \in \mathbb{R}^{784 \times 10}$ has 10 channels. To illustrate the underlying memory patterns, we plot $\sqrt{L}\rho$ for each channel in Figure 6. We observe that for the decorrelated sMNIST dataset, the underlying memory function exhibits a sudden spike at a long distance, implying the curse of memory when $\Re(w) < 0$. This observation is confirmed in Figure 5 (Right), which shows that initializing $\Re(w) = 0$ outperforms the case with a negative real part. We also apply our methods to the Long Range Arena (LRA) benchmark (Tay et al., 2021), which features six diverse tasks ranging from text to image processing. Given that we lack precise knowledge of the memory function for each LRA task, we opt to initialize a fraction of the real part as zero and compare this setup to the default S4D model (Gu et al., 2022c). In particular, for each single layer of an $L$-layer S4D model, which has a feature dimension of $d$ and a state size of $m$, there are $d$ state vectors $w \in \mathbb{C}^m$. At initialization, a fraction $p \in [0, 1]$ of

| | ListOps | Text | Retrieval | Image | Pathfinder | PathX | Avg |
|---|---|---|---|---|---|---|---|
| Baseline | 60.47 | 86.18 | 89.46 | 88.19 | 93.06 | 91.95 | 84.89 |
| Initialize 10% of $\Re(w)$ to be 0 | **61.44** | **88.05** | **90.73** | **89.11** | **95.58** | **97.55** | **87.08** |
| Ratio for $\Re(w) \geq 0$ after training | 1.29% | 1.99% | 2.58% | 4.31% | 4.31% | 4.04% | 3.09% |

Table 1: Test accuracy for training S4D on the LRA benchmark with different fractions of zero real part at initialization.

these state vectors is randomly selected to have their real parts set to zero. When $p = 0$, the training proceeds following the baseline setup. As $p$ increases, the model starts with more zero real parts. To ensure credible results, we exclude any reparameterization method on the zero real part, allowing the model to adapt from the data during training. This approach isolates the impact of the zero real part on performance without confounding variables introduced by reparameterization. All models were trained with a 6-layer architecture, maintaining the original S4D training conditions as specified in the work by Gu et al. (2022c). We present both the test accuracy and the ratio of *non-negative* real part parameters to the total $Ldm$ real part parameters upon completion of training in Table 1. We can see that, initializing an appropriate fraction of state vectors with zero real parts enables the model to outperform the default S4D configuration. Importantly, even post-training, some non-negative real parts persist, suggesting that the model retains stability and effectively adapts to the data. We provide more experimental details in Appendix B. In Appendix B.2, we add ablation studies on the gray-sCIFAR dataset with varied fractions $p$ of zero real part at initialization.

### 3.3 IMAGINARY PART INDUCES AN APPROXIMATION-ESTIMATION TRADEOFF

In the previous subsection, we show that the real part $\Re(w)$ is related with the long-term memory when training SSMs. In this subsection, we focus on the imaginary part $\Im(w)$. We will demonstrate how $\Im(w)$ influences the conditioning of the SSM optimization problem within a convex framework. Additionally, from an approximation standpoint, we reveal an approximation-estimation tradeoff that arises when training SSMs with a particular class of target functions.

**Convergence analysis.** Here we consider the continuous-time SSM (2) and assume that the read-out vector $c$ is in $\mathbb{R}^m$. This real-value assumption is necessary in the current version to get an theoretical estimate for the spectrum of the induced Gram matrix because we use the Gershgorin circle theorem to prove Theorem 3.4. This theorem is applicable only when the matrix has dominant diagonal entries. If the vector $c$ is complex-valued, the resulting Gram matrix would not be diagonal-dominant, rendering the Gershgorin circle theorem ineffective. Suppose the true input-output relation is given by some real-valued target function $\rho^*(s) \in L^1[0,\infty) \wedge L^2[0,\infty)$ with $y^*(t) = \int_0^t \rho^*(s)x(t-s)ds$. We use the squared difference between the SSM output $y(t)$ and the target output $y^*(t)$ at some terminal time $T > 0$ averaged over input distributions, which can be written as

$$\mathcal{L}(c,a) := \mathbb{E}_x \left( y(T) - y^*(T) \right)^2. \tag{4}$$

To make the theoretical analysis amenable, we assume that $x(t)$ is sampled from white noise, i.e., $x(T-s)ds = dW_s$ where $W_s$ is the canonical real-valued Wiener process. Then by Itô's isometry (Proposition C.2), the expected risk (4) can be rewritten as $\mathcal{L}(c,w) = \int_0^T \left( c^\top \Re(e^{ws}) - \rho^*(s) \right)^2 ds$. In the practical training, the sequence length is very long and thus we take $T \to \infty$ to investigate the effect of long-term memory. To study the effects of the state vector initialization, we consider the following convex optimization problem where $w$ is fixed.

$$\underset{c \in \mathbb{R}^m}{\arg\min} \, \mathcal{L}_c := \int_0^\infty \left( \sum_{j=1}^m c_j \Re(e^{w_j s}) - \rho^*(s) \right)^2 ds. \tag{5}$$

From the perspective of function approximation, the HiPPO framework (Gu et al., 2020) initializes $w$ such that the SSM basis kernel functions $\{\Re(e^{w_j s})\}_{j=1}^\infty$ are orthogonal in $L^2[0,\infty)$ w.r.t. some measure $\omega(s)$. In this paper, we discover the effects of the state initialization on the optimization problem (5). Let $c^*$ be one of the solution of the convex problem (5), then $c^*$ is a stationary point that satisfies $Gc^* = \int_0^\infty \Re(e^{ws})\rho^*(s)ds$, where $G \in \mathbb{R}^{m \times m}$ is a Gram matrix with

$$[G]_{j,k} = \int_0^\infty \Re(e^{w_j s})\Re(e^{w_k s})ds. \tag{6}$$

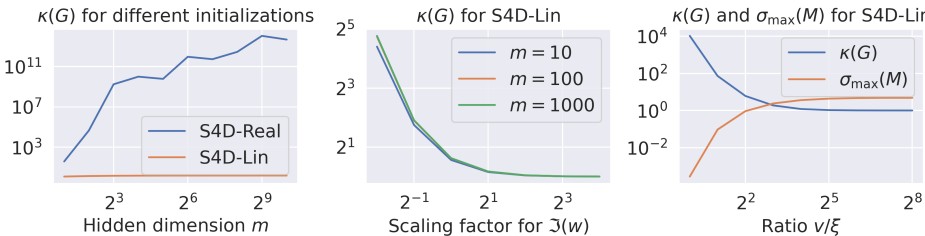

Figure 7: (Left) Condition number $\kappa(G) := \frac{\lambda_{\max}(G)}{\lambda_{\min}(G)}$ for S4D-Real and S4D-Lin with different hidden size $m$. (Middle) $\kappa(G)$ for S4D-Lin with different $m$ by varying scaling factors of the imaginary part $\Im(a)$. (Right) $\kappa(G)$ and approximation measure $\sigma_{\max}(M)$ (in the approximation-estimation tradeoff part) for S4D-Lin by different ratios of model frequencies $v$ and target frequencies $\xi$.

Therefore, the spectrum of the Gram matrix $G$ determines the numerical stability and convergence rate of optimization algorithms for solving the convex problem (5). We show in the following proposition that when $w \in \mathbb{R}^m$ and all $w_j$ are distinct, or when $w \in \mathbb{C}^m$ and all the imaginary parts $\Im(w)$ are non-zero and distinct, then $G$ is positive definite.

**Proposition 3.3.** *Let $w_j = a_j + i \cdot v_j$ with $a_j, v_j \in \mathbb{R}$ for $j = 1, \ldots, m$. If all $v_j = 0$, i.e., $w \in \mathbb{R}^m$, then $G$ is positive definite given that all $a_j$ are distinct. If $v_j$ are all non-zero, i.e., $w \in \mathbb{C}^m$, then $G$ is positive definite given that all $v_j$ are distinct.*

The proof is based on the argument of Vandermonde matrix, and we provide details in Appendix E. Given that the gram matrix $G$ is positive-definite, we are ready to study its spectrum. In the following theorem, we show that for complex-valued SSMs, the gram matrix $G$ can be well-conditioned provided that the imaginary parts $\Im(w)$ are well separated.

**Theorem 3.4.** *Let $\lambda_{\min}(G), \lambda_{\max}(G)$ be the extreme eigenvalues of $G$ defined in (6), and let $\coth(x) = \frac{e^{2x}+1}{e^{2x}-1}$. Suppose that $w_j = -0.5 + i \cdot v_j$ for $v_j \in \mathbb{R}$, and we define the separation distance $\delta := \min_{j \neq k} |v_j - v_k|$. Then if $\delta > 0$, we have*

$$1.19 - \frac{3\pi}{4\delta} \coth\left(\frac{\pi}{\delta}\right) < \lambda_{\min}(G) \leq \lambda_{\max}(G) < \frac{5}{12} + \frac{3\pi}{4\delta} \coth\left(\frac{\pi}{\delta}\right).$$

The proof is based on the Gershgorin circle theorem, with details provided in Appendix F. The setup $w_j = -0.5 + i \cdot v_j$ follows the configurations in Gu et al. (2022a;c; 2023). This theorem shows that the Gram matrix $G$ can be well-conditioned when the separation distance $\delta$ is large. One example is that for the commonly used S4D-Lin initialization (Gu et al., 2022c), $v_j = \pi \cdot j$. Then the separation distance $\delta = \pi$. Numerical calculations show that $0.2 < \lambda_{\min}(G) \leq \lambda_{\max}(G) < \sqrt{2}$, meaning that $G$ is well-conditioned for any hidden size $m$, and its condition number has a uniform upper bound w.r.t. $m$. Note that $x \coth(x) \geq 1$ and is increasing on $[0, \infty)$, which implies that the bound for $\lambda_{\min}(G)$ is non-trivial when $\frac{3\pi}{4\delta} \coth\left(\frac{\pi}{\delta}\right) < 1.19$. By numerically solving this inequality, it is sufficient to have $\delta > 2.3$. However, Proposition 3.3 suggests that as long as $\delta > 0$, the positive-definiteness of $G$ is guaranteed. This indicates a gap between the lower bound and the actual minimal eigenvalue, which we leave for future research.

**Real vs complex.** We can now compare real-valued SSMs and complex-valued SSMs in terms of the conditioning of the convex optimization problem (5), which is determined by the condition number of $G$. For real-valued SSMs with the S4D-Real initialization (Gu et al., 2022c), where $w_j = -j$, we have $G_{j,k} = \frac{1}{j+k}$. In this case, $G$ is a Hilbert matrix, whose condition number grows exponentially with respect to its size $m$ (Todd, 1953). For complex-valued SSMs with $w_j = -0.5 + i v_j$, Theorem 3.4 indicates that if the separation distance $\delta$ remains uniformly large with respect to $m$, then $G$ can be well-conditioned even for larger values of $m$. For S4D-Lin initialization, we already know that $0.2 < \lambda_{\min}(G) \leq \lambda_{\max}(G) < \sqrt{2}$ by the above argument. Therefore, unlike real-valued SSMs, the condition number of $G$ in the complex-valued case can remain well-conditioned even for large $m$, given that the imaginary parts are well separated. This difference is illustrated in Figure 7 (Left), where we compare the exact condition numbers for S4D-Real and S4D-Lin. As the scaling factor

of the imaginary part increases, the separation distance also increases. Figure 7 (Middle) shows that the Gram matrix $G$ for S4D-Lin becomes better conditioned, validating Theorem 3.4.

**Approximation-estimation tradeoff.** Despite the fact that complex-valued SSMs with adequately separated imaginary parts $\Im(w)$ enhance the conditioning of $G$, we cannot simply initialize $w$ with widely separated $\Im(w)$. This is because $\Im(w)$ determines the frequencies that the SSM can capture, and misaligned frequencies relative to the target $\rho^*$ lead to a large approximation error $\mathcal{L}_{c^*}$. For example, suppose that the target memory function $\rho^*(s) = e^{-s/2}\hat{c}^\top \cos(\xi s)$ with $\hat{c}, \xi \in \mathbb{R}^m$. Let $w = -0.5 + iv$ for $v \in \mathbb{R}^m$, then we have

$$\mathcal{L}_{c^*} = \int_0^\infty \rho^{*2}(s)ds - \left(\int_0^\infty e^{-\frac{s}{2}}\cos(vs)\rho^*(s)ds\right)^\top G^{-1}\left(\int_0^\infty e^{-\frac{s}{2}}\cos(vs)\rho^*(s)ds\right) = \hat{c}^\top M c,$$

where $M \in \mathbb{R}^{m\times m}$ is given by

$$\int_0^\infty e^{-s}\cos(\xi s)\cos(\xi s)^\top ds - \left(\int_0^\infty e^{-s}\cos(\xi s)\cos(vs)^\top ds\right) G^{-1}\left(\int_0^\infty e^{-s}\cos(vs)\cos(\xi s)^\top ds\right).$$

We can see that the maximum singular value $\sigma_{\max}(M)$ of $M$ determines the approximation error. Now, let's consider a limiting case when $v_j = \mu j$ with $\mu \to \infty$. According to Lemma C.5, we know that $G = \frac{1}{2}\mathbb{I}_m$, a scaled identity matrix, possesses the best possible conditioning. Furthermore, if $\xi$ is finite, then as $\mu \to \infty$, $\int_0^\infty e^{-s}\cos(v_j s)\cos(\xi_k s)\,ds = 0$, indicating that the worst approximation error $\int_0^\infty \rho^{*2}(s)\,ds$. On the other hand, if we aim to minimize the approximation error, we might align the frequencies such that $v = \xi$. However, when the target function comprises closely spaced frequencies $\xi_1, \ldots, \xi_m$, such alignment may cause $G$ to have a large condition number (as per Theorem 3.4). Balancing these two aspects reveals an approximation-estimation tradeoff, which is crucial when selecting an SSM initialization. Numerical evidence for this tradeoff is illustrated in Figure 7 (Right). In this figure, we set $\xi_j = 0.1\pi j$ with a relatively small separation distance $\delta = 0.1\pi$, and we vary the ratio $v_j/\xi_j$ from $2^0$ to $2^8$. As the ratio increases, the optimization is expected to improve, while the approximation deteriorates. This trend is shown in Figure 7 (Right), where the induced Gram matrix $G$ becomes better-conditioned, whereas the approximation measure $\sigma_{\max}(M)$ increases. In practice for a specific task, if we manage to accurately recover the memory function from the sequential data, we can then apply a Fourier transform to identify the dominant frequencies of the memory function. Given a state size $m$, we can greedily select $m$ frequencies that exhibit the largest separation distance from these dominant frequencies to initialize the imaginary part. Our theory suggests that this approach achieves an optimal balance between the tradeoff of approximation and estimation. However, practically, recovering the memory function accurately from the data is challenging, and hyperparameter tuning might be needed to find the optimal balance.

## 4    CONCLUSION

In this paper, we study the question proposed in the Introduction section, focusing on two initialization schemes for state space models (SSMs): the timescale $\Delta$ and the state matrix $W$. Regarding the timescale $\Delta$, we investigate it from the perspective of training stability at initialization. Our findings indicate that its dependency on sequence length is determined by data autocorrelation. By analyzing data autocorrelation, we can initialize $\Delta$ to enhance SSM training for tasks involving fixed-length sequences. For the state matrix $W$, we differentiate between the real part $\Re(W)$ and the imaginary part $\Im(W)$. The real part $\Re(W)$ is crucial for capturing long-term memory in temporal data. Allowing for a zero real part can effectively mitigate the curse of memory while maintaining training stability at initialization, provided the timescale is appropriately initialized. The imaginary part $\Im(W)$ affects the conditioning of the SSM optimization problem. A well-separated $\Im(W)$ leads to a well-conditioned Gram matrix, improving the convergence rate. However, from an approximation standpoint, excessively increasing the separation distance can result in a frequency mismatch between the SSM and the target function, leading to an approximation-estimation tradeoff. These three components are intricately linked as a *data-dependent* initialization scheme for SSMs. There are several potential future interesting directions. For instance, we have not discussed the effects of gating (Mehta et al., 2023) and model depth on the approximation and optimization of SSMs, which we leave for future research.

## 5 ACKNOWLEDGEMENT

We would like to thank the anonymous reviewers for their constructive comments. Q. Li is supported by the National Research Foundation, Singapore, under the NRF fellowship (project No. NRF-NRFF13-2021-0005).

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

## A    RELATED WORKS

**Optimization of SSMs.** Recurrent-based architectures are known for two issues: training stability and computational cost (Pascanu et al., 2013). To mitigate these challenges and capture long range dependencies more effectively in sequence modeling, the S4 model was introduced with novel parameterization, initialization, and discretization techniques (Gu et al., 2022b). Recent updates to the S4 model have further simplified the hidden state matrix by using a diagonal matrix, thereby improving computational efficiency (Gu et al., 2022c; Gupta et al., 2022; Orvieto et al., 2023). Additionally, regularization methods such as dropout, weight decay, and data-dependent regularizers (Liu & Li, 2024) are employed with SSMs to prevent overfitting. In this study, we explore how temporal dependencies in input sequences impact initialization schemes in terms of optimization, with a particular focus on the timescale and state matrix.

**Curse of memory in SSMs.** The "curse of memory" is a newly introduced concept that highlights the difficulty recurrent-based models face in capturing long-term memory (Li et al., 2021; 2022), and has been discussed in recent works (Cirone et al., 2024; Sieber et al., 2024; Zucchet & Orvieto, 2024). This issue arises due to the exponential decay property of the model's kernel basis functions. A common strategy to parameterize the real part of the state matrix's eigenvalues involves stable parameterization (Gu et al., 2022c; Wang & Li, 2024), ensuring stable training dynamics even if the input sequence is infinitely long. However, this stable parameterization constrains the real part of the state matrix's eigenvalues to be strictly negative, thereby limiting the model's ability to capture long-term memory. In this paper, we argue that if input sequences have fixed lengths, it is reasonable to set the real part of the eigenvalues to zero by appropriately setting the timescale. This relaxation allows the model to capture long-term memory while still maintaining training stability.

## B    EXPERIMENTS DETAILS

In this section, we provide more experiment details that produce Figure 1, 2, 3, 4, 5, 6 and Table 1 in section 3.

**Figure 1 (Left).** The synthetic dataset that we use to produce Figure 1 (Left) is i.i.d. sampled from standard normal distribution with dimension 128, i.e., each input sequence is of shape $(1, L, 128)$ where $L$ is its sequence length. We use a ZOH discretized diagonal SSM layer (3) with hidden size $m = 32$, model dimension $d = 128$ to handle the 128 dimensional dataset. We initialize the state vector $w$ by S4D-Lin with real part $-0.5$. The read-out vector $c$ is initialized as i.i.d. standard normal distribution. We vary $\Delta_{\min}$ and $\Delta_{\max}$ in the SSM layer and use the Adam optimizer (Kingma, 2014) to train the hyperparmeters $\Delta, \Re(w), \Im(w), C$ without weight decay. The learning rate for $\Delta, \Re(w), \Im(w)$ is 0.001 and the learning rate for $c$ is 0.1.

**Figure 1 (Middle), 2, 3.** The synthetic datasets that we use to produce these figures are Gaussian processes with mean zero and varied autocovariance functions $\mathbb{E}[x_i x_j] = K(i, j)$ for $i, j = 1, 2, \ldots, L$. Specifically, the 'iid' dataset refers to $K(i, j) = \delta_{i-j}$; the 'ou' dataset refers to $K(i, j) = \exp(-|i - j|/2)$; the 'rbf' dataset refers to $K(i, j) = \exp(-\pi|i - j|^2)$; and the autocovariance matrix for the 'rand' dataset is given by $\Sigma\Sigma^\top/L$ where $\Sigma \in \mathbb{R}^{L \times L}$ is a random matrix with i.i.d. entries sampled from a uniform distribution $U[0, \sqrt{3}]$. For all the four synthetic datasets, we have $K(i, i) = 1$. The plot for Figure 1 (Middle) records the maximal eigenvalue of the sample matrix that we fix the data size to be 1000 and vary the sequence length $L$ as plotted. So we can see some deviations between theory and practice. For Figure 2 & 3, we also use the 1-dimensional SSM layer (3) with S4D-Lin initialization on $\Im(w)$ and vary the real part $\Re(w)$ to be $-0.5$ or 0.

**Figure 1 (Right), 4, 5 (Right), 6.** For the resized sequential image datasets, we choose to resize the original images with resize rates $[0.5, 1, 1.5, 2, 2.5, 3, 3.5, 4]$. Then we standardize the whole images and flatten them into 1-d sequence. For sequential MNIST (sMNIST) dataset, the sequence length is $784r^2$ and for sequential CIFAR10 (sCIFAR10), the sequence length is $1024r^2$ where $r$ is the resize rate. The plot for the maximal eigenvalue of the autocorrelation matrix and the output value are based on the whole training set. We use the 1-dimensional SSM layer (3) with S4D-Lin initialization and vary the real part $\Re(w)$ to be $-0.5$ or 0 to calculate the output value magnitude. For the decorrelated sMNIST dataset, we choose the original MNIST dataset and the decorrelation transformation is given by a centered matrix with a whitening matrix after flattening images. The centered matrix is the mean of the sequential data along the batch dimension, and the whitening

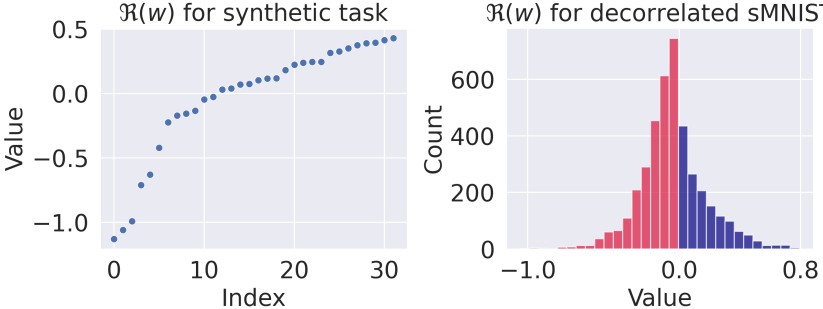

Figure 8: Behavior of the real part $\Re(w)$ after training on the synthetic task and the decorrelated sMNIST dataset.

matrix has shape $L \times L$. The whitening matrix can be obtained by SVD on the data matrix. To train the decorrelated sMNIST dataset, we use a 128-dimensional SSM layer (3) with $m = 32$ and GELU activation (Hendrycks & Gimpel, 2016) on the model output, and also apply a gated linear unit after the GELU activation. The experiment for $\Re(w) = -0.5$ follows the default training setup in Gu et al. (2022c), and for the experiment with $\Re(w) = 0$, we initialize all the real part $\Re(w)$ to be zero without any reparameterization method. We use dropout with rate 0.1 and apply a decoder layer for classification. We use Adam optimizer with learning rate 0.001 on $\Delta, \Re(w), \Im(w)$ and AdamW optimizer with weight decay 0.01 on the rest hyperparameters. For the plot of the memory function in Figure 6, we solve a least square problem by taking the pseudo inverse of the sequence matrix $X \in \mathbb{R}^{50000 \times 784}$ and then get the recovered memory function $\rho$.

**Figure 5 (Left), (Middle).** The comparisons on zero real part and negative real part in Figure 5 (Left) & (Middle) are conducted on a 1-dimensional synthetic dataset. We sample the training and test dataset from i.i.d. standard normal distribution with length 128. The training sample size and the test sample size are both 1000. We use the SSM layer (3) with $m = 32$, S4D-Lin initialization on $\Im(w)$ and initialize the timescale $\Delta = 1/\sqrt{128}$. We use Adam optimizer with learning rate 0.001 on $\Delta, \Re(w), \Im(w)$ and learning rate 0.01 on $c$.

**Table 1.** Compared with the default training setup in Gu et al. (2022c) for S4D models, the only difference is that we randomly select a fraction $p \in [0, 1]$ over the feature dimension such that the selected state vectors are initialized with zero real part. For these selected vectors, their corresponding timescale $\Delta_0 \in \mathbb{R}^{p \cdot d}$ is initialized to a constant. As suggested by Theorem 3.1, to ensure the stability, $\Delta_0$ can be chosen to be $\mathcal{O}(1/\sqrt{L\lambda_{\max}(\mathbb{E}[xx^\top])})$. Here we do not conduct a prior numerical check on the spectrum of the data autocorrelation, so we simply take an upper bound of $\lambda_{\max}(\mathbb{E}[xx^\top])$, which is given by $\mathcal{O}(L)$ if we have normalized the sequences such that $\mathbb{E}[\|x\|^2] = 1$. In that case, $\Delta_0 = \mathcal{O}(1/L)$. In the default training setup (Gu et al., 2022c), the model timescale $\Delta$ is sampled from a uniform distribution $U[\Delta_{\min}, \Delta_{\max}]$ with $\Delta_{\min} \sim \mathcal{O}(1/L)$. Hence, for the LRA benchmark we simply initialize $\Delta_0$ as a constant $\Delta_{\min}$ as in Gu et al. (2022c).

We also summarize the in other hyperparameters for training S4D on the LRA benchmark in Table 2. And we provide the behavior of the real part $\Re(w)$ after training on the synthetic task (ref Figure 5 (Left), (Middle)) and the decorrelated sMNIST dataset (ref Figure 5 (Right)) in Figure 8. We can see that with zero real part at initialization, it is possible that there remains some non-negative real part after training if no reparameterization method is introduced. But the improvement on the experiments indicates that the model can learn from the data effectively even without constraining the real part values.

## B.1 ADDITIONAL EXPERIMENTS FOR S4D-LEGS INITIALIZATION

In this subsection, we include more experiment results in Figure 9, 10, 11, 12 for SSMs with S4D-Legs (Gu et al., 2022c) initialization on the imaginary part $\Im(w)$. The S4D-Legs initialization is an approximation on the original S4-Legs initialization (Gu et al., 2022b) by taking diagonal part of the diagonal plus low-rank HiPPO-Legs matrix. In Figure 9, 10, 11, we plot the magnitude of

| | $D$ | $H$ | $N$ | Dropout | Learning rate | Batch size | Epochs | Weight decay |
|---|---|---|---|---|---|---|---|---|
| ListOps | 6 | 256 | 4 | 0 | 0.01 | 32 | 40 | 0.05 |
| Text | 6 | 256 | 4 | 0 | 0.01 | 16 | 32 | 0.05 |
| Retrieval | 6 | 256 | 4 | 0 | 0.01 | 64 | 20 | 0.05 |
| Image | 6 | 512 | 64 | 0.1 | 0.01 | 50 | 200 | 0.05 |
| Pathfinder | 6 | 256 | 64 | 0.0 | 0.004 | 64 | 200 | 0.05 |
| PathX | 6 | 256 | 64 | 0.0 | 0.0005 | 16 | 50 | 0.05 |

Table 2: List of the S4D model hyperparameters for the LRA benchmark, where $D, H, N$ denote the depth, feature dimension and hidden state space dimension respectively.

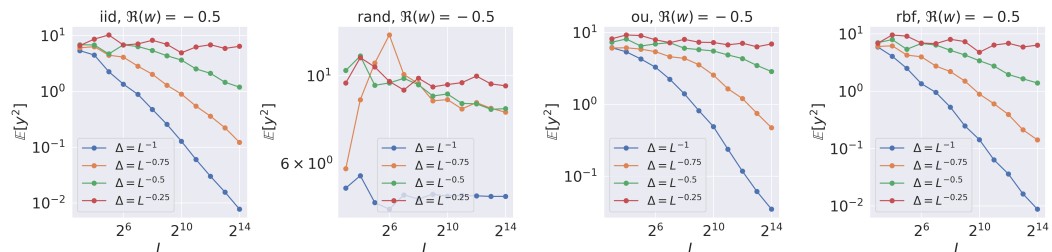

Figure 9: The expected magnitude of the SSM output value on synthetic sequences with S4D-Legs initialization and different autocorrelation. The real part $\Re(w) = -0.5$ follows the common practice and we consider four dependencies between the timescale $\Delta$ and the sequence length $L$.

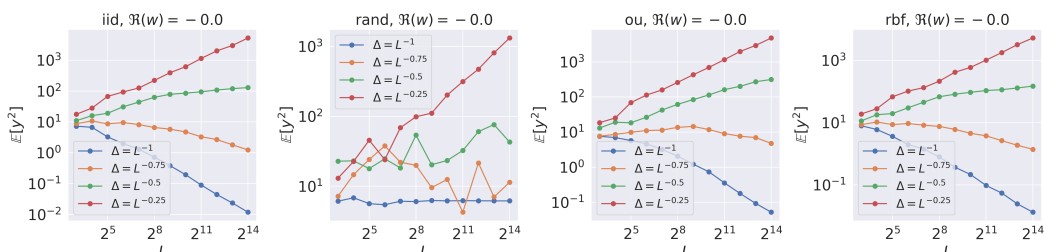

Figure 10: The expected magnitude of the SSM output value on synthetic sequences with S4D-Legs initialization and different autocorrelation and different dependencies between $\Delta$ and $L$. The real part $\Re(w)$ is set to be zero.

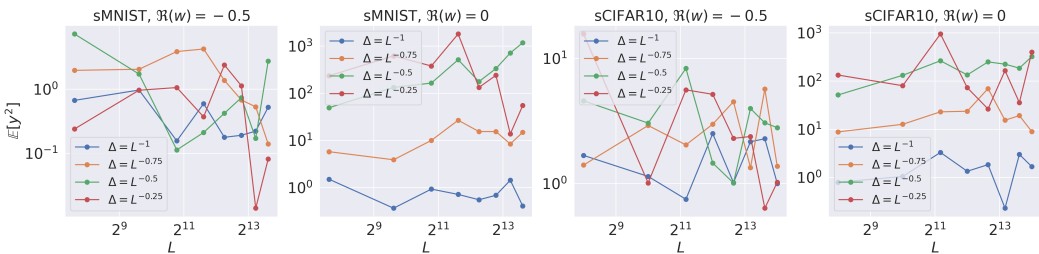

Figure 11: The expected magnitude of the SSM output value for S4D-Legs initialization on sequential image datasets with different resize rates (ranging from 0.5 to 4) and different dependencies between $\Delta$ and $L$.

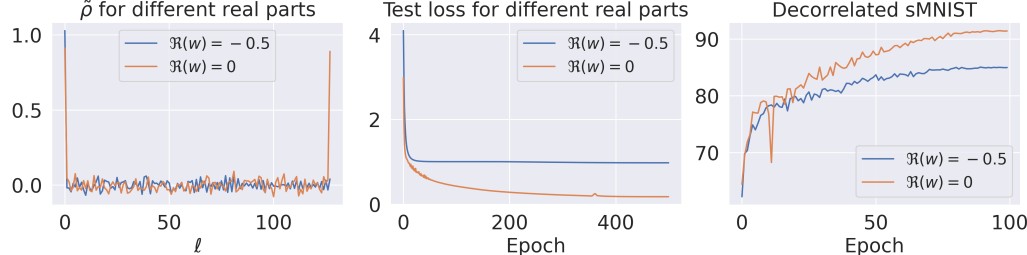

Figure 12: (Left) Training a diagonal SSM (3) with S4D-Legs initialization on a task that requires long-term memory. The learned memory function $\tilde{\rho}$ effectively captures the spike in long-range dependencies. However, it struggles to do so when the real part is negative. (Middle) Test loss on the long-term memory task when initializing $\Re(w) = 0$ and $\Re(w) = -0.5$. (Right) Test accuracy for training a diagonal SSM with S4D-Legs initialization on decorrelated sequential MNIST dataset with different real parts at initialization.

| Initialize a fraction $p$ of $\Re(w)$ to be 0 | $p = 0$ | $p = 0.1$ | $p = 0.2$ | $p = 0.3$ | $p = 0.4$ | $p = 0.5$ |
|---|---|---|---|---|---|---|
| Accuracy | 84.09 (0.47) | **84.60** (0.38) | 84.23 (0.49) | 83.77 (0.46) | 83.50 (0.42) | 83.19 (0.39) |
| Ratio for $\Re(w) \geq 0$ after training | 0% | 3.90% | 7.62% | 10.85% | 14.34% | 17.25% |

Table 3: Test accuracy for training a 4-layer S4D model in sCIFAR dataset with varied fractions of zero real part at initialization.

the SSM output value given the S4D-Legs initialization for both zero real part and negative real part cases. The experiment settings follow the guidelines we introduce before with only a change on the initialization of $\Im(w)$. We can see that for S4D-Legs initialization, our conclusion still holds in the sense that negative real part is stable at initialization for all all the scaling that we considered in this paper, while for zero real part, the dependencies of $\Delta$ on $L$ varies for different sequence autocorrelation. We also compare the effects of real parts on optimization with S4D-Legs initialization. The results are shown in Figure 12 and we obtain consistent results as the S4D-Lin initialization. One interesting finding is that on the decorrelated sMNIST dataset, the comparison between Figure 5 (Right) and Figure 12 (Right) shows that the S4D-Lin initialization outperforms the S4D-Legs initialization in both zero real part and negative real part cases.

### B.2 ABLATION STUDIES ON THE EFFECTS OF FRACTIONS OF ZERO REAL PART AT INITIALIZATION

In this subsection, we conduct ablation studies on the gray-sCIFAR dataset (with sequence length 1024) to evaluate the benefit of initializing the real part to zero in multi-layer S4D models, while making minimal modifications. Based on our theory, zeroing the real part can alleviate the *curse of memory* in scenarios where the memory function exhibits a long memory pattern. But since we do not have a precise knowledge of the memory function for the gray-sCIFAR dataset, we do ablation studies on the effects of zero real part by varying the fraction $p$ as specified in Section 3.2. For the selected state vectors with zero real part, their corresponding timescale $\Delta_0 \in \mathbb{R}^{p \cdot d}$ is initialized as a constant 0.001. We use a 4-layer S4D model with feature dimension 128 and vary $p$ across $[0, 0.1, 0.2, 0.3, 0.4, 0.5]$. We present both the test accuracy and the ratio of non-negative real part parameters to the total $4 \times 32 \times 128 = 16384$ real part parameters upon completion of training in Table 3. We can see that initializing an appropriate fraction of state vectors with zero real parts enables the model to outperform the default S4D configuration. Importantly, even post-training, some non-negative real parts persist, suggesting that the model retains stability and effectively adapts to the data.

## C AUXILIARY LEMMAS

In this section, we provide the description for Itô's isometry and a few auxiliary lemmas that we will need for the proofs of Theorem 3.1, Proposition 3.3 and Theorem 3.4.

**Lemma C.1.** *If $\Re(z) \leq 0$, then*

$$\left| \frac{e^z - 1}{z} \right| \leq 1.$$

*Proof.* Notice that

$$\frac{|e^z - 1|}{|z|} = \frac{\left| \int_0^z e^s ds \right|}{|z|} \leq \frac{\int_0^z |e^s||ds|}{|z|} = \frac{\int_0^z e^{\Re(z)}|ds|}{|z|} \leq \frac{\int_0^z |ds|}{|z|} = 1,$$

which finishes the proof. □

**Lemma C.2** (Itô's isometry). *Let $W : [0,T] \times \Omega \to \mathbb{R}$ denote the canonical real-valued Wiener process defined up to time $T > 0$, and let $X : [0,T] \times \Omega \to \mathbb{R}$ be a stochastic process that is adapted to the natural filtration of the Wiener process. Then*

$$\mathbb{E}\left[ \left( \int_0^T X_t \, dW_t \right)^2 \right] = \mathbb{E}\left[ \int_0^T X_t^2 \, dt \right],$$

*where $\mathbb{E}$ denotes expectation with respect to classical Wiener measure.*

**Lemma C.3** (Gershgorin circle theorem). *Let $A$ be a complex $n \times n$ matrix, with entries $a_{ij}$. For $i \in \{1, \ldots, n\}$, let $R_i$ be the sum of the absolute value of the non-diagonal entries in the $i$-th row: $R_i = \sum_{j \neq i} |a_{ij}|$. Let $D(a_{ii}, R_i) \subseteq \mathbb{C}$ be a closed disc centered at $a_{ii}$ with radius $R_i$. Then every eigenvalue of $A$ lies within at least one of the discs $D(a_{ii}, R_i)$.*

**Lemma C.4.** *For any $t \in \mathbb{R}$,*

$$\sum_{n=1}^{\infty} \frac{1}{n^2 + t^2} = -\frac{1}{2t^2} + \frac{\pi}{2t} \coth(\pi t).$$

*Proof.* This is a side result of the Basel problem. The related proof can be found in the Wiki page. We omit it here. □

**Lemma C.5.** *For any $v_j, v_k \in \mathbb{R}$, we have*

$$\int_0^{\infty} e^{-s} \cos(v_j s) \cos(v_k s) ds = \frac{1}{2} \left( \frac{1}{1 + (v_j - v_k)^2} + \frac{1}{1 + (v_j + v_k)^2} \right).$$

*Proof.* Notice that

$$
\begin{aligned}
&\int_0^{\infty} e^{-s} \cos(v_j s) \cos(v_k s) ds \\
=& \frac{1}{2} \int_0^{\infty} e^{-s} \cos((v_j - v_k)s) ds + \frac{1}{2} \int_0^{\infty} e^{-s} \cos((v_j + v_k)s) ds \\
=& \frac{1}{2} \int_0^{\infty} \Re\left( \exp\left( -s + i \cdot (v_j - v_k)s \right) \right) ds + \frac{1}{2} \int_0^{\infty} \Re\left( \exp\left( -s + i \cdot (v_j + v_k)s \right) \right) ds \\
=& \frac{1}{2} \Re\left( \frac{1}{1 - i \cdot (v_j - v_k)} + \frac{1}{1 - i \cdot (v_j + v_k)} \right) \\
=& \frac{1}{2} \left( \frac{1}{1 + (v_j - v_k)^2} + \frac{1}{1 + (v_j + v_k)^2} \right).
\end{aligned}
$$

□

**Lemma C.6** (Hanson-Wright inequality). *Let $X = (X_1, \ldots, X_n) \in \mathbb{R}^n$ be a random vector with independent components $X_i$ which satisfy $\mathbb{E}X_i = 0$ and $\|X_i\|_{\psi_2} \leq K$. Let $A$ be an $n \times n$ matrix. Then, for every $t \geq 0$,*

$$\mathbb{P}\left\{ |X^\top A X - \mathbb{E}X^\top A X| > t \right\} \leq 2 \exp\left[ -c \min\left( \frac{t^2}{K^4 \|A\|_F^2}, \frac{t}{K^2 \|A\|} \right) \right],$$

where $c$ is a positive absolute constant and the subgaussian norm $\| \cdot \|_{\psi_2}$ is defined as

$$\|\xi\|_{\psi_2} = \sup_{p \geq 1} p^{-1/2} (\mathbb{E}|\xi|^p)^{1/p}.$$

*In particular, if $\xi$ is a standard normal distribution, then $\|\xi\|_{\psi_2} = \sqrt{8/3}$.*

*Proof.* We refer the proof to Rudelson & Vershynin (2013). $\qquad\square$

## D   PROOF OF THEOREM 3.1

In this section, we prove the upper bound on the second moment of the model output value in Theorem 3.1.

*Proof.* First, we may express the model output $y_L$ in a matrix form. To do so, we rewrite $c$ as a $2m \times 1$ vector $(\Re(c_1), \ldots, \Re(c_m), \Im(c_1), \ldots \Im(c_m))^\top$ that contains the real and imaginary part of $c$, and let $V$ to be a $2m \times L$ Vandermonde-like matrix

$$V := \begin{bmatrix} \Re\left(\frac{e^{\Delta w_1}-1}{\Delta w_1} e^{\Delta w_1 0}\right) & \Re\left(\frac{e^{\Delta w_1}-1}{\Delta w_1} e^{\Delta w_1 1}\right) & \cdots & \Re\left(\frac{e^{\Delta w_1}-1}{\Delta w_1} e^{\Delta w_1 (L-1)}\right) \\ \vdots & \vdots & & \vdots \\ \Re\left(\frac{e^{\Delta w_m}-1}{\Delta w_m} e^{\Delta w_m 0}\right) & \Re\left(\frac{e^{\Delta w_m}-1}{\Delta w_m} e^{\Delta w_m 1}\right) & \cdots & \Re\left(\frac{e^{\Delta w_m}-1}{\Delta w_m} e^{\Delta w_m (L-1)}\right) \\ -\Im\left(\frac{e^{\Delta w_1}-1}{\Delta w_1} e^{\Delta w_1 0}\right) & -\Im\left(\frac{e^{\Delta w_1}-1}{\Delta w_1} e^{\Delta w_1 1}\right) & \cdots & -\Im\left(\frac{e^{\Delta w_1}-1}{\Delta w_1} e^{\Delta w_1 (L-1)}\right) \\ \vdots & \vdots & & \vdots \\ -\Im\left(\frac{e^{\Delta w_m}-1}{\Delta w_m} e^{\Delta w_m 0}\right) & -\Im\left(\frac{e^{\Delta w_m}-1}{\Delta w_m} e^{\Delta w_m 1}\right) & \cdots & -\Im\left(\frac{e^{\Delta w_m}-1}{\Delta w_m} e^{\Delta w_m (L-1)}\right) \end{bmatrix}.$$

Then $y_L$ can be written in a matrix form

$$y_L = \Delta \cdot c^\top V J x,$$

where $J \in \mathbb{R}^{L \times L}$ is a row reversed identity matrix, i.e.

$$J = \begin{pmatrix} 0 & \cdots & 0 & 1 \\ 0 & \cdots & 1 & 0 \\ \vdots & \ddots & \vdots & \vdots \\ 1 & \cdots & 0 & 0 \end{pmatrix}.$$

Furthermore, we may connect $V$ with a standard Vandermonde matrix $V_L$, by noticing that

$$\Phi V = D V_L,$$

where $V_L$ is a $2m \times L$ complex Vandermonde matrix with $2m$ nodes $e^{\Delta w_1}, e^{\Delta \bar{w}_1}, \ldots, e^{\Delta w_m}, e^{\Delta \bar{w}_m}$:

$$V_L = \begin{pmatrix} 1 & e^{\Delta \bar{w}_1} & \cdots & e^{\Delta \bar{w}_1 (L-1)} \\ \vdots & \vdots & \cdots & \vdots \\ 1 & e^{\Delta \bar{w}_m} & \cdots & e^{\Delta \bar{w}_m (L-1)} \\ 1 & e^{\Delta w_1} & \cdots & e^{\Delta w_1 (L-1)} \\ \vdots & \vdots & \cdots & \vdots \\ 1 & e^{\Delta w_m} & \cdots & e^{\Delta w_m (L-1)} \end{pmatrix} \in \mathbb{C}^{2m \times L},$$

$\Phi$ is a scaled unitary matrix

$$\Phi := \left( \begin{array}{cccc|cccc} 1 & 0 & \cdots & 0 & i & 0 & \cdots & 0 \\ 0 & 1 & \cdots & 0 & 0 & i & \cdots & 0 \\ \vdots & \vdots & \ddots & \vdots & \vdots & \vdots & \ddots & \vdots \\ 0 & 0 & \cdots & 1 & 0 & 0 & \cdots & i \\ \hline 1 & 0 & \cdots & 0 & -i & 0 & \cdots & 0 \\ 0 & 1 & \cdots & 0 & 0 & -i & \cdots & 0 \\ \vdots & \vdots & \ddots & \vdots & \vdots & \vdots & \ddots & \vdots \\ 0 & 0 & \cdots & 1 & 0 & 0 & \cdots & -i \end{array} \right) \in \mathbb{C}^{2m \times 2m}.$$

with $\Phi\Phi^H = \Phi^H\Phi = 2\mathbb{I}_{2m}$, and $D$ is a diagonal matrix

$$
D = \begin{pmatrix}
\frac{e^{\Delta\bar{w}_1}-1}{\Delta\bar{w}_1} & & & & & \\
& \ddots & & & & \\
& & \frac{e^{\Delta\bar{w}_m}-1}{\Delta\bar{w}_m} & & & \\
& & & \frac{e^{\Delta w_1}-1}{\Delta w_1} & & \\
& & & & \ddots & \\
& & & & & \frac{e^{\Delta w_m}-1}{\Delta w_m}
\end{pmatrix} \in \mathbb{C}^{2m\times 2m}.
$$

Hence, we have $V = \frac{1}{2}\Phi^H DV_L$. Notice that both $\Re(\Delta w_j)$ and $\Re(\Delta\bar{w}_j)$ are non-positive, then by Lemma C.1 we have $\|D\| \le 1$. Now combining it with $V = \frac{1}{2}\Phi^H DV_L$ and the fact that the exchange matrix $J$ is an orthogonal matrix, then when $\Re(w_j) \le 0$ for all $j$, we have

$$
\begin{aligned}
\mathbb{E}_{c,x}[y_L^2] &= \mathbb{E}_{c,x}\left[\left(\Delta \cdot c^\top V J x\right)^2\right] \\
&= \Delta^2 \mathbb{E}_c\left[c^\top V J \mathbb{E}_x[xx^\top] JV^\top c\right] \\
&\le \frac{\Delta^2}{2}\mathbb{E}_c[\|c\|^2]\lambda_{\max}(\mathbb{E}[xx^\top])\lambda_{\max}(V_L V_L^H) \\
&\le \frac{\Delta^2 m}{2}\lambda_{\max}(\mathbb{E}[xx^\top])\operatorname{Tr}(V_L V_L^H) \\
&= \Delta^2 m\lambda_{\max}(\mathbb{E}[xx^\top])\sum_{j=1}^{m}\left(\left(e^{\Delta\Re(w_j)}\right)^0 + \cdots + \left(e^{\Delta\Re(w_j)}\right)^{L-1}\right) \\
&= \Delta^2 m^2 L\lambda_{\max}(\mathbb{E}[xx^\top]),
\end{aligned}
$$

which finishes the proof. $\qquad\square$

It is also possible to derive a high-probability bound using Lemma C.6 (the Hanson-Wright inequality). It's important to note that we do not make any assumptions about the input sequential data; instead, we only assume that the read-out vector $c$ is i.i.d. Gaussian, as stated in Theorem 3.1. This allows us to apply the high-probability bound to the expression $c^\top V J \mathbb{E}_x[xx^\top]JV^\top c$, where $V J \mathbb{E}_x[xx^\top]JV^\top$ is a deterministic matrix. By applying the Hanson-Wright inequality in Lemma C.6, we take a $\delta > 0$, and let $A = V J \mathbb{E}_x[xx^\top]JV^\top$, $K = \sqrt{8/3}$, then by solving $\frac{t}{K^2\|A\|} = \frac{\log(2/\delta)}{c}$, we have $t = \frac{8\log(2/\delta)}{3c}\|A\|$. Then for small enough $\delta$, i.e., for large enough $t$, we have $\frac{t^2}{K^4\|A\|_F^2} > \frac{t}{K^2\|A\|}$. Therefore, by Lemma C.6 we get with probability at least $1 - \delta$,

$$
\begin{aligned}
\mathbb{E}_x[y_L^2] &\le \Delta^2 m^2 L\lambda_{\max}(\mathbb{E}[xx^\top]) + \frac{8\Delta^2\log(2/\delta)}{3c}\|A\| \\
&\le \Delta^2 m^2 L\lambda_{\max}(\mathbb{E}[xx^\top]) + \frac{4\Delta^2\log(2/\delta)}{3c}\lambda_{\max}(\mathbb{E}[xx^\top])\operatorname{Tr}(V_L V_L^H) \\
&\lesssim \Delta^2 mL\lambda_{\max}(\mathbb{E}[xx^\top])\left(m + \frac{\log(1/\delta)}{c}\right),
\end{aligned}
$$

where $\lesssim$ hides a positive absolute constant.

## E    PROOF OF PROPOSITION 3.3

In this section, we show the proof for Proposition 3.3.

*Proof.* Since $G_{j,k} = \int_0^\infty \Re(e^{w_j s})\Re(e^{w_k s})ds$, then for any $\xi \in \mathbb{R}^m$, we have

$$
\xi^\top G\xi = \int_0^\infty \left(\sum_{j=1}^m \xi_j\Re(e^{w_j s})\right)^2 ds \ge 0.
$$

Hence, $G$ is positive semi-definite for both real-valued $w$ and complex-valued $w$. Let $\xi^\top G\xi = 0$, then $\sum_{j=1}^{m} \xi_j \Re(e^{w_j s}) = 0$ for $s \geq 0$.

When $a \in \mathbb{R}^m$, we take the discrete time points $s = 0, 1, \ldots, m$ to form $m$ equations. Note that $\Re(e^{w_j s}) = e^{w_j s}$. If $w_j$ are distinct, then the Vandermode matrix given by $w_1, \ldots, w_m$ is invertible, indicating that the only solution for $\sum_{j=1}^{m} \xi_j \Re(e^{w_j s}) = 0$ is $\xi_j = 0$ for $j = 1, \ldots, m$. Thus, $G$ is positive definite in that case.

When $w \in \mathbb{C}^m$ with distinct imaginary parts, we can always find a scaling factor $\gamma > 0$ such that $e^{\gamma w_1}, \ldots, e^{\gamma w_m}, e^{\gamma \bar{w}_1}, \ldots, e^{\gamma \bar{w}_m}$ are distinct, where $\bar{w}$ is the conjugate of $w$. Then by the argument of Vandermonde matrix, the only solution of the equation $\sum_{j=1}^{m} \xi_j e^{w_j s} + \sum_{j=1}^{n} \hat{\xi}_j e^{\bar{w}_j s} = 0$ for $s \geq 0$ is that $\xi_j = \hat{\xi}_j = 0$ for $j = 1, \ldots, m$. Since $2\Re(e^{w_j s}) = e^{w_j s} + e^{\bar{w}_j s}$, then $\sum_{j=1}^{m} \xi_j \Re(e^{w_j s}) = 0$ only has zero solution.

Combining these two cases we finish the proof. $\qquad\square$

## F  PROOF OF THEOREM 3.4

In this section, we prove Theorem 3.4 based on the Gershgorin circle theorem (Lemma C.3).

*Proof.* First, we need to bound both the diagonal entry and the off-diagonal sum. The diagonal entry $G_{j,j} = \frac{1}{2}(1 + \frac{1}{1+4v_j^2})$, which can be bounded as

$$\frac{1}{2}\left(1 + \frac{1}{1 + 4v_j^2}\right) \leq G_{j,j} \leq 1, \quad j = 1, \ldots, m.$$

For the off-diagonal sum, we have $\forall j = 1, \ldots, m$,

$$
\begin{aligned}
2R_j &= 2\sum_{k \neq j} |G_{j,k}| \\
&= \sum_{k \neq j} \frac{1}{1 + (v_j - v_k)^2} + \sum_{k \neq j} \frac{1}{1 + (v_j + v_k)^2} \\
&< \sum_{k=1}^{\infty} \frac{2}{1 + \delta^2 k^2} + \sum_{k=1}^{\infty} \frac{1}{1 + (v_j + v_k)^2} - \frac{1}{1 + 4v_j^2} \\
&< \sum_{k=1}^{\infty} \frac{2}{1 + \delta^2 k^2} + \sum_{k=1}^{\infty} \frac{1}{1 + v_j^2 + v_k^2} - \frac{1}{1 + 4v_j^2} \\
&< \sum_{k=1}^{\infty} \frac{2}{1 + \delta^2 k^2} + \sum_{k=0}^{\infty} \frac{1}{1 + v_j^2 + \delta^2 k^2} - \frac{1}{1 + 4v_j^2} \\
&= \frac{2}{\delta^2}\sum_{k=1}^{\infty} \frac{1}{1/\delta^2 + k^2} + \frac{1}{\delta^2}\sum_{k=1}^{\infty} \frac{1}{(1 + v_j^2)/\delta^2 + k^2} + \left(\frac{1}{1 + v_j^2} - \frac{1}{1 + 4v_j^2}\right),
\end{aligned}
$$

where the first inequality is due to the fact that the minimal separation distance $\min_{j \neq k} |v_j - v_k| \geq \delta$, and the last inequality is because $v_j > 0$ and reordering $\{v_k\}_{k \geq 1}$ does not affect the result for $\sum_{k=1}^{\infty} \frac{1}{1+v_j^2+v_k^2}$. Then by Lemma C.4, we have

$$
\begin{aligned}
\frac{2}{\delta^2}\sum_{k=1}^{\infty} \frac{1}{1/\delta^2 + k^2} + \frac{1}{\delta^2}\sum_{k=1}^{\infty} \frac{1}{(1 + v_j^2)/\delta^2 + k^2} &< \frac{3}{\delta^2}\sum_{k=1}^{\infty} \frac{1}{1/\delta^2 + k^2} \\
&= \frac{3}{\delta^2}\left(-\frac{\delta^2}{2} + \frac{\pi\delta}{2}\coth\left(\frac{\pi}{\delta}\right)\right) \\
&= -\frac{3}{2} + \frac{3\pi}{2\delta}\coth\left(\frac{\pi}{\delta}\right).
\end{aligned}
$$

Hence we have,

$$
\begin{aligned}
G_{j,j} - R_j &> \frac{1}{2}\left(1 + \frac{1}{1 + 4v_j^2}\right) - \frac{1}{2}\left(-\frac{3}{2} + \frac{3\pi}{2\delta}\coth\left(\frac{\pi}{\delta}\right)\right) - \frac{1}{2}\left(\frac{1}{1 + v_j^2} - \frac{1}{1 + 4v_j^2}\right) \\
&> \frac{5}{4} - \frac{1}{2}\max\left(\frac{1}{1 + x^2} - \frac{2}{1 + 4x^2}\right) - \frac{3\pi}{4\delta}\coth\left(\frac{\pi}{\delta}\right) \\
&> 1.19 - \frac{3\pi}{4\delta}\coth\left(\frac{\pi}{\delta}\right).
\end{aligned}
$$

Under the same argument, we get

$$
\begin{aligned}
G_{j,j} + R_j &< 1 + \frac{1}{2}\left(-\frac{3}{2} + \frac{3\pi}{2\delta}\coth\left(\frac{\pi}{\delta}\right)\right) + \frac{1}{2}\max\left(\frac{1}{1 + v_j^2} - \frac{1}{1 + 4v_j^2}\right) \\
&< \frac{1}{4} + \frac{3\pi}{4\delta}\coth\left(\frac{\pi}{\delta}\right) + \frac{1}{2}\max\left(\frac{1}{1 + v_j^2} - \frac{1}{1 + 4v_j^2}\right) \\
&= \frac{5}{12} + \frac{3\pi}{4\delta}\coth\left(\frac{\pi}{\delta}\right).
\end{aligned}
$$

Combining the two bounds and Lemma C.3, we finish the proof. $\qquad\square$

