# OpenReview forum: "Autocorrelation Matters: Understanding the Role of Initialization Schemes for State Space Models"
_ICLR.cc/2025/Conference — ICLR 2025 Poster_

### Official Review · Reviewer_BxH3 · 2024-10-21

**Soundness:** 3
**Presentation:** 3
**Contribution:** 3
**Rating:** 6
**Confidence:** 4

**Summary:**

This paper provides a theoretical analysis of a state-space model from three perspectives: discretisation size ($\Delta$), the real components of the diagonal weights, and their imaginary counterparts. It discusses the scaling laws of $\Delta$, highlights how zero real parts contribute to long memory, and introduces an approximation-estimation tradeoff influenced by the imaginary components.

**Strengths:**

**(I)** The analyses presented are rigorous, with clear explanations connecting the results to different initialization strategies, which is crucial for understanding SSMs.

**(II)** The paper is well-structured, with illustrative examples that help to clarify the theoretical concepts.

**Weaknesses:**

**(I)** The paper primarily focuses on theoretical aspects, with limited empirical evaluation beyond basic synthetic examples. It would be beneficial to include practical applications that demonstrate the utility of the proposed methods.

**(II)** The explanation of the three theoretical components sounds a bit separated. While they all fall under the umbrella of initializations, a stronger connection would give the paper a more cohesive narrative.

**(III)** Although the theoretical results are clearly presented, they are dense. More emphasis on distilling key messages and offering practical guidelines would enhance the paper's accessibility.

**Questions:**

**(I)** The analysis focuses on the single-input single-output case. How would these results extend to the multiple-input multiple-output (S5) case? High-level insights would suffice.

**(II)** The paper assumes ZOH discretisation. How would the results differ with bilinear discretisation?

**(III)** Several questions regarding Theorem 4.1:

   (a) The upper bound applies to the expected output. Is there a corresponding probabilistic bound?

   (b) Can the upper bound be shown to be tight?

   (c) How does the real part of $w_j$ affect this bound? Specifically, how would increasing or decreasing it impact the upper bound?

   (d) The analysis focuses on the final output. How would the results change if the sum of outputs $\sum_{\ell = 1}^L y_\ell^2$  were considered instead? This pooling is adopted by many SSMs.

**(IV)** The paper explores various settings for how $\Delta$ should scale with $L$. Given a specific task, what method would you use to determine the appropriate $\Delta$?

**(V)** When the real part of $w$ is set to zero at initialization, is it still trained in practice? If so, do you enforce non-positivity? In this case, how do you handle reparameterization, especially since you can’t use the logarithm of it? Could you also show the behavior of the real part of $w$ after training? If it becomes negative, what is the benefit of zero initialization?

**(VI)** On line 376, the assumption is made that $c$ is real-valued. However, many SSM implementations use a complex $c$. What necessitates this assumption?

**(VII)** Given a specific task, how would you determine the appropriate balance between approximation and estimation tradeoff? Would this be driven by hyperparameter tuning or derived from theoretical insights?

---

> ### Author Response · Authors · 2024-11-23
> **Authors' response to Reviewer BxH3.**
>
> We thank the reviewer for the valuable comments and suggestions. In the following, we address the weakness and questions raised.
>
> Weakness 1. *It would be beneficial to include practical applications that demonstrate the utility of the proposed methods.*
>
>   * We appreciate this suggestion.
>   In light of your comment, we have expanded our empirical evaluation to include the Long Range Arena (LRA) benchmark, which provides a robust set of tasks to validate our approach under practical conditions.
>   Specifically, we trained 6-layer S4D models on this benchmark, initializing 10% of the real part to zero and configuring the timescale for this component based on Theorem 3.1.
>   Our findings, detailed in the second table of the **General Responses**, show that our method consistently outperforms the default S4D model across all tasks, underscoring its practical effectiveness.
>   For more detailed experimental results and analysis, we invite the reviewer to consult the **General Responses**.
>   **We have also included the LRA benchmark results in Section 3.2, Appendix B, and Appendix B.2 of the revised manuscript to provide a more comprehensive evaluation of our approach.**
>
>
> Weakness 2 \& 3. A stronger connection among the theoretical components and more emphasis on distilling key messages would enhance the paper's accessibility.
>
>   * Thank you for the helpful feedback.
>   In response, we have elaborated on the interconnections among the three theoretical components in the **General Responses**.
>   The central message is that these components collectively form a cohesive, data-dependent initialization scheme for S4D models. Specifically, given an input sequential dataset, we can estimate its autocorrelation to initialize the model timescale $\Delta$.
>   This allows us to harness the memory function of the input-output mapping to set the real part $\Re(w)$, which helps address the curse of memory challenges associated with long sequences.
>   Lastly, by accurately recovering the memory function from the data, we can balance the approximation-estimation tradeoff by initializing the imaginary part $\Im(w)$ based on the dominant frequencies of the memory function.
>   This structured approach enhances the model's efficacy and aligns theoretical insights with practical application.
>   **To make the paper more accessible, we have expanded on these connections in the Introduction and Conclusion sections of the revised manuscript.**

---

> > ### Author Response · Authors · 2024-11-23
> > **Authors' response to Reviewer BxH3 (continue).**
> >
> > Q1. *How would these results extend to the multiple-input multiple-output (S5) case?*
> >
> >   * For a multiple-input multiple-output (MIMO) model, the output is essentially a linear combination of several single-input single-output (SISO) models.
> >   As a result, we can extend our theory to the MIMO scenario by examining each SISO model individually along with its respective input-output mapping.
> >   This approach allows us to apply our theoretical framework to the MIMO case seamlessly, leveraging the understanding of individual SISO model dynamics to inform the broader MIMO model behavior.
> >  **In light of the reviewer's feedback, we have included a discussion on extending our insights to the MIMO case in Remark 2.1 of the revised manuscript.**
> >
> > Q2. *How would the results differ with bilinear discretization?*
> >
> >   * This is an interesting question. Different discretization methods lead to different matrix forms for the model's input-output mapping.
> >   In this paper, we concentrate on the Zero-Order Hold (ZOH) discretization method, which is widely used in practice.
> >   Under ZOH discretization, the model's input-output mapping involves Vandermonde matrices, as discussed in the proof of Theorem 3.1.
> >   The spectrum of these matrices is well-explored in existing literature.
> >   On the other hand, bilinear discretization results in mapping that involves bilinear terms, which require more advanced techniques to yield theoretical insights.
> >   Extending our theory to accommodate bilinear discretization presents an intriguing opportunity for future research.
> >   **We have included the above discussion in Remark 2.1 of the revised manuscript.**
> >
> > Q3. *Several questions regarding Theorem 3.1.*
> >
> >   * The upper bound applies to the expected output. Is there a corresponding probabilistic bound?
> >
> >     * Yes, we can derive a high-probability bound using the Hanson-Wright inequality.
> >     It's important to note that we do not make any assumptions about the input sequential data; instead, we only assume that the read-out vector $c$ is i.i.d. Gaussian, as stated in Theorem 3.1.
> >     This allows us to apply the high-probability bound to the expression $c^\top V J \mathbb{E}_x[x x^\top] J V^\top c$, where $V J \mathbb{E}_x[x x^\top] J V^\top$ is a deterministic matrix.
> >     By applying the Hanson-Wright inequality directly, we obtain a high-probability bound for the model output $\mathbb{E}_x[y_L^2]$.
> >     **The exact bound can be found in the end of Appendix D of the revised manuscript.**
> >
> >   * Can the upper bound be shown to be tight? How does the real part of $w_j$ affect this bound?
> >
> >     * Our upper bound is applicable for all cases where $\Re(w) \leq 0$.
> >     As the real part $\Re(w)$ approaches zero, the exponential decay rate of the SSM kernel slows, resulting in a tighter bound. Specifically, when the real part is zero and the input data lacks temporal dependency, the bound becomes tight up to a constant factor. This occurs because the data autocorrelation matrix $\mathbb{E}_x[x x^\top]$ simplifies to a diagonal matrix under these conditions.
> >     **In light of the reviewer's question, we have included the end of Theorem 3.1 in the revised manuscript.**
> >
> >   * How would the results change if the sum of outputs $\sum_{\ell = 1}^L y_\ell^2$ were considered instead?.
> >
> >     * The stability result presented in Theorem 3.1 remains unaffected by the average-pooling operation.
> >     It's important to note that $\mathbb{E}[\frac{1}{L} \sum_{\ell=1}^L y_\ell^2] \leq \frac{1}{L} \sum_{\ell=1}^L \Delta^2 m^2 \ell \lambda_{\max} (\mathbb{E}[x x^\top]) \leq \Delta^2 m^2 L \lambda_{\max} (\mathbb{E}[x x^\top])$. Thus, we obtain the same bound as specified in Theorem 3.1.
> >     However, altering the output mode can result in a markedly different memory pattern for the input-output mapping, necessitating a case-by-case analysis.
> >     We believe that exploring which output mode is optimal for various tasks, particularly in relation to the *curse of memory*, is a compelling direction for further research.
> >     **We have included the above discussion in Remark 3.2 of the revised manuscript.**

---

> ### Author Response · Authors · 2024-11-23
> **Authors' response to Reviewer BxH3 (continue).**
>
> Q4. *The paper explores various settings for how $\Delta$ should scale with $L$. Given a specific task, what method would you use to determine the appropriate $\Delta$?*
>
>   * Given a specific task with a training sequential dataset, we can estimate the maximal eigenvalue of the data autocorrelation matrix $\mathbb{E}\_x[x x^\top]$.
>   Then we can initialize the model timescale $\Delta$ as $1/\sqrt{L \lambda\_{\max}(\mathbb{E}\_x[x x^\top])}$ to ensure the model's stability as specified in Theorem 3.1.
>   **In light of the reviewer's feedback, we have included the above discussion in the end of Theorem 3.1 in the revised manuscript.**
>
> Q5. *When the real part of $w$ is set to zero at initialization, is it still trained in practice? If so, do you enforce non-positivity? In this case, how do you handle reparameterization, especially since you can’t use the logarithm of it? Could you also show the behavior of the real part of $w$ after training? If it becomes negative, what is the benefit of zero initialization?*
>
>   * Yes, the real part is trained but *without* any reparameterization method.
>   This approach means $\Re(w)$ is likely to become positive during training.
>   Our goal is to allow the model to learn directly from the data without introducing new variables, such as reparameterization methods, which could complicate the analysis.
>   Otherwise, it would be unclear whether the improvements are due to the zero real part or the introduced reparameterization method.
>   The experimental results showcased in the **General Responses** illustrate that even with a portion of the real part initialized to zero, the model successfully learns from the data, and some real parts of the state vector become non-negative after training.
>   In light of the reviewer's question, we have added a discussion on why not constraining the parameter values during training in Section 3.2 of the revised manuscript.
>   **For the behavior of the real part of $w$ after training, we provide a figure (Figure 8) in Appendix B of the revised manuscript, which shows the distribution of the real part of $w$ after training on the synthetic task and the sMNIST dataset.**
>   **For the LRA benchmark, we show the ratio of non-negative real parts to the total real parts in Table 1 and 3 in the revised manuscript.**
>
> Q6. *The assumption for $c$ is real-valued. However, many SSM implementations use a complex $c$. What necessitates this assumption?*
>
>   * In the current version of our work, this assumption is necessary because we utilize the Gershgorin circle theorem to bound the eigenvalues of the Gram matrix, as explained in the proof of Theorem 3.3.
>   This theorem is applicable only when the matrix has dominant diagonal entries.
>   If the vector $c$ is complex-valued, the resulting Gram matrix would not be diagonal-dominant, rendering the Gershgorin circle theorem ineffective.
>   However, we believe that this assumption could be relaxed by employing alternative techniques to bound the spectrum of the Gram matrix, and we plan to explore this possibility in future work.
>   **In light of the reviewer's feedback, we have included the above discussion in Section 3.3 of the revised manuscript.**
>
> Q7. *Given a specific task, how would you determine the appropriate balance between approximation and estimation tradeoff? Would this be driven by hyperparameter tuning or derived from theoretical insights?*
>
>   * For a specific task, if we manage to accurately recover the memory function from the sequential data, we can then apply a Fourier transform to identify the dominant frequencies of the memory function.
>   Given a state size $m$, we can greedily select $m$ frequencies that exhibit the largest separation distance from these dominant frequencies to initialize the imaginary part. Our theory suggests that this approach achieves an optimal balance between the tradeoff of approximation and estimation.
>   However, practically, recovering the memory function accurately from the data is challenging, and hyperparameter tuning might be needed to find the optimal balance.
>   **In light of the reviewer's question, we have included the above discussion in the end of Section 3.3 of the revised manuscript.**

---

> > ### Comment · Reviewer_BxH3 · 2024-11-26
> >
> > Thank you for the thorough response and detailed explanations. I have a few follow-up points for clarification and further exploration:
> >
> > (I) The explanation of a MIMO system as a "linear combination" of SISO systems remains somewhat unclear. Could you elaborate on this? Since MIMO systems differ in input and output dimensions from SISO systems, the notion of "linear combination" here seems ambiguous.
> >
> > (II) Regarding the observation that the real parts of $w$ are likely positive after training, does this not suggest that the system may become unstable or exhibit unbounded growth as the sequence extends? It would be interesting to understand if there are implicit mechanisms, perhaps connected to the $\Delta$ parameter, that mitigate such behavior.
> >
> > (III) The relationship between the real and imaginary components also raises intriguing questions. For instance, when you set a portion of real parts to be zero in your additional experiments, what are the corresponding imaginary parts? Will that make a difference in the model performance?
> >
> > Overall, this work represents a valuable contribution. My recommendation for acceptance remains unchanged. However, the discussions surrounding the three main aspects of the work still feel somewhat decoupled and only loosely connected by the overarching data-oriented objective. Exploring these interconnections further could enhance the narrative, though this may be better suited for future work.

---

> > > ### Author Response · Authors · 2024-11-26
> > > **Authors' response to Reviewer BxH3's follow-up questions.**
> > >
> > > We thank the reviewer for the follow-up questions. In the following, we address the three questions raised.
> > >
> > > Q1. *More elaborations on how to extend to MIMO systems.*
> > >
> > >   * Let's consider the following MIMO system with input dimension $d$ and output dimension $\hat{d}$,
> > >   $$
> > >   \frac{dh(t)}{dt} = W h(t) + B x(t), \quad y(t) = \Re(C h(t)),
> > >   $$
> > >   where $x(t) = (x_1(t),\ldots,x_d(t))^\top \in \mathbb{R}^{d}, y(t) = (y_1(t),\ldots,y_{\hat{d}}(t))^\top\in \mathbb{R}^{\hat{d}}, W \in \mathbb{C}^{m \times m}, h(t) \in \mathbb{C}^m, C = (c_1,\ldots,c_{\hat{d}})^\top \in \mathbb{C}^{\hat{d}\times m}$.
> > >   Following the SISO setup we assume the read-in matrix $B \in \mathbb{R}^{m \times d}$ to be an all-one matrix, and $W$ to be a diagonal matrix with diagonal entries $w \in \mathbb{C}^m$.
> > >   Then
> > >   $$
> > >   \frac{dh(t)}{dt} = W h(t) + B x(t) = W h(t) + b \sum_{j=1}^d x_j(t),
> > >   \quad
> > >   y_{k} = \Re(c_k^\top h(t)), k = 1,\ldots,\hat{d},
> > >   $$
> > >   where $b$ is an all-one vector in $\mathbb{R}^m$.
> > >   Therefore, each entry $y_k(t)$ of the output vector $y(t)$ is the sum of $d$ SISO models w.r.t. the single input $x_1(t),\ldots,x_d(t)$ and the corresponding read-out vector $c_k$.
> > >
> > > Q2. *Positive real parts will induce unstable systems for unbounded sequence length*
> > >
> > >   * Yes indeed. In this paper, we focus on fixed-length tasks, or more precisely, the maximal sequence length of test/inference data does not exceed the maximal training sequence length.
> > >   If we want to do test/inference on longer sequences, we need to adjust the timescale accordingly.
> > >   It would be interesting to investigate how to adapt the timescale for longer sequences, and we plan to explore this in future work.
> > >
> > > Q3. *What are the imaginary parts for zero real parts?*
> > >
> > >   * To make minimal modifications on the default S4D model, we only change the real part and the corresponding timescale.
> > >   For example, if the baseline training follows S4D-Legs initialization, then the imaginary parts for zero real parts also follow the S4D-Legs initialization.
> > >   If the baseline training follows S4D-Lin initialization, then the imaginary parts for zero real parts also follow the S4D-Lin initialization.
> > >   We do not compare the performance of different initialization methods when both the real and imaginary parts vary, which will make the comparison more complicated.
> > >
> > > We hope these responses address the reviewer's questions and concerns. We appreciate the insightful questions and are grateful for the opportunity to clarify these points.
> > > We agree with the reviewer that exploring more detailed interconnections among the three initialization components is a promising direction for future research, and we plan to investigate this in our future work.

---

### Official Review · Reviewer_vKbW · 2024-11-01

**Soundness:** 4
**Presentation:** 3
**Contribution:** 3
**Rating:** 8
**Confidence:** 2

**Summary:**

An analysis of the effects of some parameters of the so-called SSM layer is provided. Specifically, the authors analyze 1) the relation between the timescale $\Delta$ and the sequence length $L$, 2) the effect of the (zero) real part of the time evolution matrix, and 3) the effect of the imaginary part of the time evolution matrix, both theoretically and empirically.

**Strengths:**

- The claims are clear, and most of them are supported both theoretically and empirically.
- The topic is indeed important as the SSM-based models are increasingly used in various problems.
- Although the analyses may not be surprising per se in terms of dynamical systems theory, their implication especially in using SSM architecture sounds new and useful.

**Weaknesses:**

The empirical evaluation is limited to synthetic data or somewhat simple benchmark data. This fact does not much diminish the value of the paper, but reporting the applicability of the proposed theory to more real-world datasets would certainly be appreciated.

A thing remained a bit unclear to me is that how the discussions are necessarily relevant only to the initialization. Doesn't it make sense to constrain the parameter values (e.g., $\Re(W)$) as guided in the given theory, not only at the initialization but also during training iterations? As the paper's title emphasizes the role of *initialization*, a bit more discussion in this regard might be helpful.

**Questions:**

No questions that may change my evaluation.

---

> ### Author Response · Authors · 2024-11-23
> **Authors' response to Reviewer vKbW.**
>
> We thank the reviewer for the valuable comments and suggestions. In the following, we address the 2 weaknesses raised.
>
> Weakness 1. *The empirical evaluation is limited to synthetic data or somewhat simple benchmark data. More real-world datasets would certainly be appreciated.*
>
>   * We appreciate the reviewer's suggestion.
>   In light of your comment, we have conducted experiments using the Long Range Arena (LRA) benchmark to further support our theory.
>   We specifically trained 6-layer S4D models on this benchmark, initializing 10% of the real part to zero and setting the timescale for this zero real part according to Theorem 3.1.
>   The results, which can be found in the second table of the **General Responses**, indicate that our method consistently surpasses the default S4D model across all tasks.
>   We encourage the reviewer to refer to the **General Responses** for more comprehensive experiment details.
>   **We have also included the LRA benchmark results in Section 3.2, Appendix B, and Appendix B.2 of the revised manuscript to provide a more comprehensive evaluation of our approach.**
>
> Weakness 2. *How the discussions are necessarily relevant only to the initialization. Doesn't it make sense to constrain the parameter values (e.g., $\Re(W)$) as guided in the given theory, not only at the initialization but also during training iterations?*
>
>   * We appreciate the reviewer's insightful question.
>   In this study, we opted not to constrain the parameter values during training.
>   This decision was made to maintain better control over the experiments and to focus on the effects of initialization schemes without introducing additional variables, such as reparameterization methods, which could complicate the analysis.
>   For instance, we initialized the real part $\Re(w)$ to zero and allowed the model to learn from the data during the training process without incorporating any reparameterization techniques.
>   Introducing such methods would make it unclear whether improvements were due to the initialization or the reparameterization itself.
>   Our results on the LRA benchmark, as detailed in the tables in the **General Responses**, demonstrate that the model can effectively learn from the data with the real part initialized to zero.
>   Even after training, some **non-negative** values remain in the state vector's real part $\Re(w)$.
>   **In light of the reviewer's feedback, we have included the above discussion on why not constraining the parameter values during training in Section 3.2 of the revised manuscript.**

---

### Official Review · Reviewer_YHUj · 2024-11-03

**Soundness:** 3
**Presentation:** 3
**Contribution:** 3
**Rating:** 6
**Confidence:** 2

**Summary:**

The paper studies the initialization strategy used in SSM from various point of view, namely the time scale, the real and imaginary part of the state matrix. It demonstrates the dependency of the SSM timescale on sequence length based on sequence autocorrelation. Further it is shown that having a zero real part for the eigenvalues of the SSM state matrix mitigates the curse of memory. Finally, the paper demonstrates that the imaginary part of the eigenvalues of the SSM state matrix determine the conditioning of SSM optimization problems, and present a approximation-estimation tradeoff when training SSMs with a specific class of target functions.

**Strengths:**

1. The paper has presented some very novel an interesting insights into the initialization scheme for SSM.
2. The analysis is well grounded and thorough.
3. The experiments are well though out, extensive and results are convincing of the claims made in the paper.

**Weaknesses:**

While the experimental results are convincing on the datasets considered in the paper, how are the expected to hold in larger scale experiments? Can authors comment on that.

What do the authors think of extending these insights or draw parallel to setting(s) where the sequence length can be varied?

**Questions:**

Please see above.

---

> ### Author Response · Authors · 2024-11-23
> **Authors' response to Reviewer YHUj.**
>
> We thank the reviewer for the valuable comments and suggestions. In the following, we address the 2 weakness raised.
>
> Weakness 1. *How are the expected results to hold in larger scale experiments? Can authors comment on that.*
>
>   * We appreciate the reviewer's insightful question.
>   In light of your comment, we conducted experiments using the Long Range Arena (LRA) benchmark to further validate our theory.
>   Specifically, we trained 6-layer S4D models on this benchmark with 10% of the real part initially set to zero.
>   Additionally, we initialized the timescale for this zero real part as outlined in Theorem 3.1. The results, presented in the second table of the **General Responses**, demonstrate that our method consistently outperforms the default S4D model across all tasks.
>   For more detailed information, we invite the reviewer to consult the **General Responses**.
>   **In light of the reviewer's feedback, we have added the LRA benchmark results to Section 3.2, Appendix B, and Appendix B.2 of the revised manuscript to provide a more comprehensive evaluation of our approach.**
>
> Weakness 2. *What do the authors think of extending these insights or draw parallel to setting(s) where the sequence length can be varied?*
>
>   * That's an excellent question.
>   Let's start by considering a scenario where the sequence length alternates between two values, $L_1$ and $L_2$.
>   To address this, we can double the model's feature dimension and initialize it separately for the first and second halves, allowing the model to accommodate two fixed-length datasets simultaneously.
>   If the sequence lengths vary even more, we could cluster them into several groups, increasing the model's feature dimension according to the same strategy.
>   Ultimately, the key idea is to expand the model's feature dimension to manage varying sequence lengths, which we believe is a promising direction for future investigation.
>   **In light of the reviewer's question, we have included a discussion on extending our insights to scenarios with varying sequence lengths in Remark 3.2 of the revised manuscript.**

---

> ### Comment · Reviewer_YHUj · 2024-11-25
>
> Dear authors,
>
> Thank you for the comments. My score remains the same

---

### Author Response · Authors · 2024-11-23
**General responses.**

We thank each reviewer for their careful reading of our work and their thoughtful feedback.
 We have revised the paper with the main changes in *red* according to the comments and suggestions.
In this unified response, first we would like to depict the whole picture and illustrate the connections among each theory in this paper.
Then we address a common concern shared by all reviewers: the absence of practical applications involving more real-world datasets.

The main goal of this paper is to provide a theoretical understanding on the effects of three specific S4D hyperparameters: the model timescale $\Delta$, the real part of the state vector $\Re(w)$, and the imaginary part of the state vector $\Im(w)$.
These components are intricately linked as a **data-dependent** initialization scheme for S4D models.
**First**, for any given sequential dataset, we can estimate its autocorrelation.
Using this information, we can apply Theorem 3.1 to initialize $\Delta$, taking into account both data autocorrelation and sequence length.
**Second**, if the true input-output mapping is represented by an underlying linear functional, often referred to as a memory function, that exhibits a long memory pattern, our second theory, detailed in Section 3.2, suggests that initializing with a zero real part $\Re(w)$ can help mitigate the challenges posed by long sequences.
**Finally**, the third theory introduced in Section 3.3 discusses an approximation-estimation tradeoff that arises when the true memory function $\rho^*$ features closely spaced frequencies.
If we can accurately recover $\rho^*$ from the sequential data, we can then initialize the imaginary part $\Im(w)$ based on the dominant frequencies of $\rho^*$, thereby finding an optimal balance informed by theoretical insights.
**To make the paper more accessible, we have elaborated on these connections in the Introduction and Conclusion sections of the revised manuscript.**

Building on the feedback from reviewers, we are expanding our experiments to include larger scale experiments.
Specifically, we are applying our first two theories to the Long Range Arena (LRA) benchmark, which features six diverse tasks ranging from text to image processing.
However, the third theory cannot be directly validated using the LRA benchmark, as it necessitates knowledge of the exact memory function for the input-output mapping—a challenge in practical scenarios.

To proceed, we begin with ablation studies on the gray-sCIFAR dataset to evaluate the benefit of initializing the real part to zero in multi-layer S4D models, while making minimal modifications. Based on our theory, zeroing the real part can alleviate the *curse of memory* in scenarios where the memory function exhibits a long memory pattern.
Given that we lack precise knowledge of the memory function for the gray-sCIFAR dataset, we opt to initialize a fraction of the real part as zero and compare this setup to the default S4D model.
In particular, for each layer of the S4D model, which has a feature dimension of $d$ and a state size of $m$, there are $d$ state vectors $w \in \mathbb{C}^m$.
At initialization, a fraction $p \in [0, 1]$ of these state vectors is randomly selected to have their real parts set to zero.
For these selected vectors, their corresponding timescale $\Delta_0 \in \mathbb{R}^{p \cdot d}$ is initialized to a constant, as suggested by our first theory.
From Figure 1 (Right), we observe that the maximal eigenvalue of $\mathbb{E}[xx^\top]$ on the sCIFAR dataset scales with $L$.
Therefore, in accordance with Theorem 3.1, we initialize $\Delta_0$ as a constant $1/L$.
Other hyperparameters are maintained as per the default S4D model settings.
When $p = 0$, the training proceeds following the baseline setup.
As $p$ increases, the model starts with more zero real parts.
To ensure credible results, we **exclude** any reparameterization strategies on the zero real part, allowing the model to adapt from the data during training.
This approach isolates the impact of the zero real part on performance without confounding variables introduced by reparameterization.
In our ablation studies, a 4-layer S4D model is used, with $p$ varied across $[0, 0.1, 0.2, 0.3, 0.4, 0.5]$.
We present both the test accuracy and the ratio of **non-negative** real part parameters to the total $Ldm$ real part parameters upon completion of training in the subsequent table.

---

> ### Author Response · Authors · 2024-11-23
> **General responses (continue).**
>
> | Initialize a fraction $p$ of $\Re(w)$ to be 0 | $p=0$ | $p=0.1$ | $p=0.2$ | $p=0.3$ | $p=0.4$ | $p=0.5$ |
> |:-----------:|:-----------:|:-----------:|:-----------:|:-----------:|:-----------:|:-----------:|
> |Accuracy| 84.09 (0.47) | **84.60** (0.38)  | 84.23 (0.49) | 83.77 (0.46) | 83.50 (0.42) | 83.19 (0.39)|
> |Ratio for $\Re(w) \geq 0$ after training| 0\% | 3.90\% | 7.62\% | 10.85\% | 14.34\% | 17.25\% |
>
> From our ablation study, we found that initializing an appropriate fraction of state vectors with zero real parts enables the model to outperform the default S4D configuration. Importantly, even post-training, some non-negative real parts persist, suggesting that the model retains stability and effectively adapts to the data.
>
> Using this setup, we proceeded to train S4D models on the entire LRA benchmark.
> For this, we initialized 10% of the real parts to zero, without further tuning the fraction $p$. Correspondingly, we set the timescale $\Delta_0$ to a constant value $\Delta_{\min}$, adhering to the default training settings.
> All models were trained with a 6-layer architecture, maintaining the original S4D training conditions as specified in the work by Gu et al. (2022).
>
> |  S4D   | ListOps          | Text            | Retrieval        | Image           | Pathfinder      | PathX           | Avg     |
> |:-----------:|:-----------:|:-----------:|:-----------:|:-----------:|:-----------:|:-----------:|:-----------:|
> |Baseline| 60.47 | 86.18  | 89.46 | 88.19 | 93.06 | 91.95| 84.89  |
> | Initialize 10\% of $\Re(w)$ to be $0$ | **61.44** | **88.05** | **90.73** | **89.11** | **95.58** | **97.55** | **87.08**|
> |Ratio for $\Re(w) \geq 0$ after training| 1.29\% | 1.99\% | 2.58\% | 4.31\% | 4.31\% | 4.04\% | 3.09\% |
>
> Given time constraints, we conducted the experiments using a single random seed.
> The experiment results demonstrate that initializing a portion of the real parts to zero consistently improves performance across all six tasks when compared to the default S4D model.
> We hope these recent findings effectively address the reviewers' concerns and would like to express our gratitude to the reviewers for their time and thoughtful consideration.
>
> **In light of the reviewers' suggestion to include more real-world datasets, we have expanded our empirical evaluation to include the Long Range Arena (LRA) benchmark in Section 3.2, Appendix B, and Appendix B.2 of the revised manuscript.**

---

### Meta-Review · Area_Chair_J4uW · 2024-12-19

**Metareview:**

The paper 'Autocorrelation Matters: Understanding the Role of Initialization Schemes for State Space Models' was reviewed by 3 reviewers who gave it an average score of 6.67 (final scores: 6+6+8). The reviewers found this work interesting, clear, and the claims supported both theoretically and empirically. All reviewers argue for accepting the paper.

**Additional Comments On Reviewer Discussion:**

The authors posted a rebuttal, and as all reviewers recommended accepting the work already in their initial reviews, there was not much discussion in the discussion phase. Nevertheless, the authors provided additional details and updates in response to the comments in the reviews.

---

### Decision · Program_Chairs · 2025-01-22

Accept (Poster)